# The fibronectin synergy site re-enforces cell adhesion and mediates a crosstalk between integrin classes

Maria Benito-Jardón[1,2], Sarah Klapproth[3], Irene Gimeno-LLuch[1,2], Tobias Petzold[4], Mitasha Bharadwaj[5], Daniel J Müller[5], Gabriele Zuchtriegel[3], Christoph A Reichel[3,6], Mercedes Costell[1,2]*

[1]Department of Biochemistry and Molecular Biology, Universitat de València, Burjassot, Spain; [2]Estructura de Recerca Interdisciplinar en Biotecnologia i Biomedicina, Universitat de València, Burjassot, Spain; [3]Walter Brendel Centre of Experimental Medicine, Ludwig-Maximilians-Universität München, Munich, Germany; [4]Medizinische Klinik und Poliklinik I, Klinikum der Universität München, Munich, Germany; [5]Eidgenössische Technische Hochschule Zürich, Basel, Switzerland; [6]Departement of Otorhinolaryngology, Ludwig-Maximilians-Universität München, Munich, Germany

*For correspondence: mercedes.costell@uv.es

Competing interests: The authors declare that no competing interests exist.

**Abstract** Fibronectin (FN), a major extracellular matrix component, enables integrin-mediated cell adhesion *via* binding of $\alpha5\beta1$, $\alpha IIb\beta3$ and $\alpha v$-class integrins to an RGD-motif. An additional linkage for $\alpha5$ and $\alpha IIb$ is the synergy site located in close proximity to the RGD motif. We report that mice with a dysfunctional FN-synergy motif (*Fn1^{syn/syn}*) suffer from surprisingly mild platelet adhesion and bleeding defects due to delayed thrombus formation after vessel injury. Additional loss of $\beta3$ integrins dramatically aggravates the bleedings and severely compromises smooth muscle cell coverage of the vasculature leading to embryonic lethality. Cell-based studies revealed that the synergy site is dispensable for the initial contact of $\alpha5\beta1$ with the RGD, but essential to re-enforce the binding of $\alpha5\beta1/\alpha IIb\beta3$ to FN. Our findings demonstrate a critical role for the FN synergy site when external forces exceed a certain threshold or when $\alpha v\beta3$ integrin levels decrease below a critical level.

## Introduction

Fibronectin (FN) is a large extracellular matrix (ECM) glycoprotein that triggers biochemical and mechanical signaling via integrin binding. FN is essential for mammalian development and tissue regeneration, and can influence disease such as cancer progression. FN is abundant in blood and in most tissues and is present in provisional matrices of healing wounds and in the stroma of tumors. FN is secreted as a disulfide-bonded dimer, assembled into fibrils of variable diameters and then crosslinked into a fibrillar network of variable rigidity (*Leiss et al., 2008*) that binds to and serves as a scaffold for numerous other ECM molecules. FN consists of three different repeating Ig-like folded units, called type I-III modules. Whereas type I and II modules are stabilized by internal disulfide bonds, the 15 type III repeats of FN lack disulfide bonds, which confers elasticity to FN fibrils and the ability to modulate fibril rigidity (*Erickson, 1994*; *Oberhauser et al., 2002*). The major cell-binding site in FN is an arginine-glycine-aspartate (RGD) motif located in the 10[th] type III module (FNIII10) that is recognized by $\alpha5\beta1$, $\alpha IIb\beta3$, and $\alpha v$-class integrins. In addition to the RGD motif, FN harbors the so-called FN synergy site in the FNIII9 module (*Obara et al., 1988*), which binds $\alpha5\beta1$

and αIIbβ3 integrins but not αv-class integrins (*Bowditch et al., 1994*). The synergy site encompasses the DRVPPSRN sequence in mouse FN and site directed mutagenesis identified the two arginine residues to be essential for all synergy site-induced functions (*Aota et al., 1994*; *Friedland et al., 2009*; *Chada et al., 2006*; *Nagae et al., 2012*).

In vitro studies have shown that the synergy site increases cell spreading (*Aota et al., 1991*), FN fibril assembly (*Sechler et al., 1997*) and platelet adhesion to FN (*Chada et al., 2006*). Based on the crystal structures, the RGD motif forms a flexible loop that physically interacts with both the α5 and β1 integrin subunits, while the synergy site contacts only the head domain of the α subunit (*Redick et al., 2000*; *Nagae et al., 2012*). The synergy site has been studied using protein- and cell-based assays, which produced different results giving rise to diverse hypotheses regarding the mechanistic properties. One hypothesis based on ultra-structural analyses of the recombinant α5β1 ectodomain and the FNIII7-10 polypeptide proposes that the synergy site aligns the binding interface of the integrin heterodimer with the RGD motif to increase the on-rate constant ($K_{on}$) of α5β1 binding to FN (*Leahy et al., 1996*; *García et al., 2002*; *Takagi et al., 2003*). A combination of theoretical and cell-based studies with FRET sensors inserted into the linker region between FNIII9 and FNIII10 concluded that cell-induced forces reversibly stretch the linker, separate the FN-RGD motif from the synergy site and switch the binding of α5β1 integrins to αv-class integrins (*Grant et al., 1997*; *Krammer et al., 2002*). Finally, using a spinning disk device, it was shown that the engagement of the synergy site allows FN-α5β1 bonds (or FN-αIIbβ3 bonds on platelets) to resist shear forces, suggesting that force exposure allows to switch the bonds from a relaxed to a tensioned state, leading to an extension of the FN-integrin bond lifetime and to adhesion strengthening (*Friedland et al., 2009*). Although these in vitro studies highlight the importance of α5β1 and αIIbβ3 integrin-binding to the synergy site, the mode of action is still unclear and the apparently important roles of these interactions have never been scrutinized in vivo using genetic loss-of-function approaches.

We decided to directly test the role of the synergy site in vivo by substituting critical residues of the FN-synergy site in mice. We report that mice carrying a homozygous inactivating mutation in the fibronectin gene (*Fn1*)-synergy site (*Fn1^{syn/syn}*) are viable, fertile, show no overt organ defects, however, display a mild bleeding tendency and delayed thrombus formation after vessel injury. The lethal intercrosses of *Fn1^{syn/syn}* mice with β3 integrin (*Itgb3*)-deficient mice and in vitro assays with purified FN^{syn} isolated from the blood of *Fn1^{syn/syn}* mice revealed three important findings: (i) the synergy site does not influence the $K_{on}$ of α5β1 integrin binding to FN-RGD, (ii) the synergy site strengthens α5β1/αIIbβ3 integrin binding to FN upon application of external force such as blood flow or internal force such as actomyosin pulling forces, and (iii) during force-induced adhesion strengthening the synergy site binding to α5β1 and αv-class integrin binding to FN compensate each other, at least in part, up to a certain force threshold.

## Results

### Normal development and prolonged trauma-induced bleeding in *Fn1^{syn/syn}* mice

To directly test the in vivo role(s) of the FN synergy site, we generated the *Fn1^{syn}* allele by substituting the two arginines ($R_{1374}$ and $R_{1379}$) of the synergy motif (DRVPPSRN) in the FN-III9 module with alanines (A) (*Figure 1A* and *Figure 1—figure supplement 1A–C*). Intercrossing of heterozygous mice (*Fn1^{+/syn}*), which showed no apparent phenotype, gave rise to homozygous offspring (*Fn1^{syn/syn}*) with a normal Mendelian ratio before and after weaning. *Fn1^{syn/syn}* mice were fertile, had normal size and weight, and aged normally. The morphology, ultrastructure, and FN distribution in heart (*Figure 1B*), liver, kidney, and lung (*Figure 1—figure supplement 1D*) were indistinguishable between *Fn1^{syn/syn}* and control littermates. Blood vessel organization in whole mount ear samples analyzed by anti-PECAM-1 and anti-αSMA immunostainings revealed no abnormalities (*Figure 1C*), and the subendothelial matrix visualized with antibodies to laminin-1, collagen IV and FN, was also normally organized in *Fn1^{syn/syn}* mice (*Figure 1—figure supplement 1E*). Altogether, these data indicate that the FN synergy site is dispensable for development and postnatal homeostasis.

Soluble plasma (p) FN is required for the stability of blood clots (*Ni et al., 2003b*). Therefore, we performed several experiments to test whether platelets require the synergy site to firmly bind FN

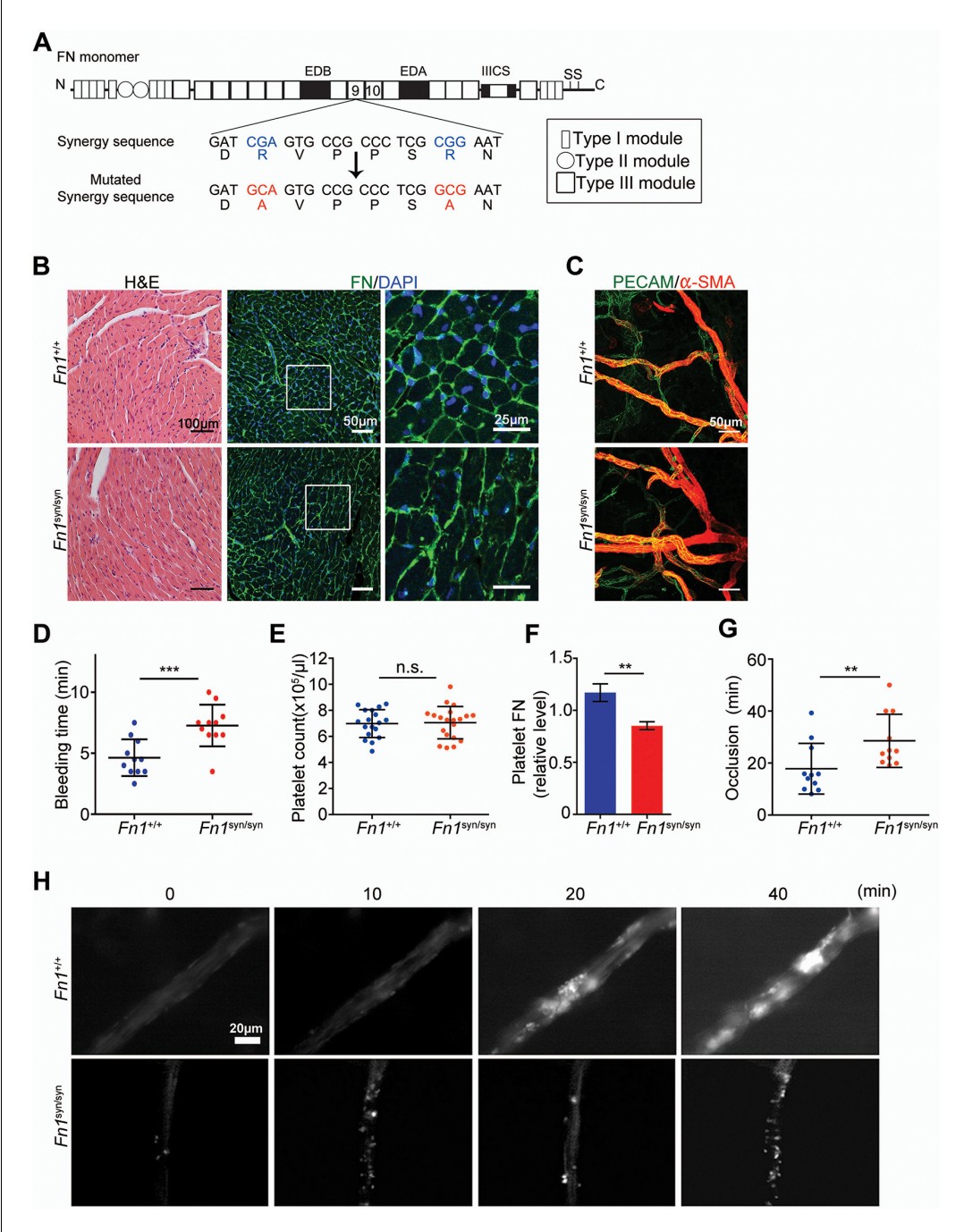

**Figure 1.** Normal tissue development and prolonged bleeding in *Fn1^syn/syn* mice. (**A**) Cartoon of FN and the nucleotide point mutations disrupting the function of the synergy site. (**B**) Representative images of 3-months-old *Fn1^+/+* and *Fn1^syn/syn* heart sections stained with H and E and immunostained for FN. (**C**) Confocal images of ear whole-mounts from 3 months-old mice immunostained with anti-PECAM-1 and anti-αSMA to visualize the dermal endothelial cell tubes and smooth muscle cells. (**D**) Bleeding time of 3-months-old *Fn1^+/+* (n = 11) and *Fn1^syn/syn* (n = 11) mice. (**E**) Platelet counts in blood samples of *Fn1^+/+* (n = 18) and *Fn1^syn/syn* (n = 19) mice. (**F**) FN content in platelets derived from *Fn1^+/+* (n = 6) and *Fn1^syn/syn* (n = 6) mice relative to their vinculin levels. (**G**) Occlusion time of injured arterioles in the cremaster muscle of 3-months-old *Fn1^+/+* (n = 11) and *Fn1^syn/syn* (n = 11) mice. (**H**) Representative still images of the arteriolar occlusion (white:platelets). Values are shown as mean ± SD; statistical significances were calculated using the Student *t*-test; **p<0.01 and ***p<0.001.

The following figure supplements are available for figure 1:

**Figure supplement 1.** Strategy used to generate the *Fn1^syn/syn* mice and tissue and platelet analysis.

*Figure 1 continued*

**Figure supplement 2.** FN levels in platelets and blood from *Fn1^syn/syn^* mice and platelet aggregation assays.

*via* their αIIbβ3 and α5β1 integrins. We measured tail bleeding time after tail biopsy (*Figure 1D*) and found a significant increase from 4.64 ± 1.50 min (mean ± SD) in *Fn1^+/+^* mice to 7.27 ± 1.71 min in *Fn1^syn/syn^* mice (p<0.001). Importantly, blood platelet counts were normal in *Fn1^syn/syn^* mice (*Figure 1E*). Since αIIbβ3 integrins mediate the uptake of pFN into platelet α-granules (*Ni et al., 2003a*), we performed Western-blotting with lysates from washed platelets and found that the FN content was significantly reduced to 70% in platelets from *Fn1^syn/syn^* mice (*Figure 1F* and *Figure 1—figure supplement 2A*), while the levels of pFN were similar in *Fn1^+/+^* (318.7 ± 24.1 µg/ml) and *Fn1^syn/syn^* (316.1 ± 31.0 µg/ml) mice (*Figure 1—figure supplement 2B*).

Importantly, plasma levels of fibrinogen were also similar in *Fn1^+/+^* (2.10 ± 0.17 mg/ml) and *Fn1^syn/syn^* (2.08 ± 0.07 mg/ml) mice. In vitro aggregation of washed platelets, induced with either collagen I, thrombin or ADP, triggered normal shape changes and aggregations (*Figure 1—figure supplement 2C–E*). To quantitatively study the velocity of thrombus formation in vivo, thrombi induction was measured in the arterioles of the cremaster muscle upon vessel injury. The experiments revealed a small delay in the onset of thrombus formation in the *Fn1^syn/syn^* mice (10.29 ± 9.04 min) that, however, was not significantly different compared to the *Fn1^+/+^* littermates (5.13 ± 3.89 min). In contrast, the time required for arteriole occlusion was significantly increased in *Fn1^syn/syn^* mice (28.56 ± 10.24 min) compared to *Fn1^+/+^* mice (17.82 ± 9.74 min) (*Figure 1G*). Notably, in 3 out of 11 *Fn1^syn/syn^* mice no total occlusion was observed after 40 min (*Figure 1H*), a defect that was never observed in control mice.

These results demonstrate that the synergy site is dispensable for development and postnatal homeostasis but is required to stabilize platelet clots in vivo and to prevent prolonged bleeding times.

## Fibroblasts delay their focal adhesion maturation on FN^syn

The assembly of FN into a fibrillar network depends on α5β1 binding to FN (*Fogerty et al., 1990*). To test whether FN assembly proceeds normally in the absence of the synergy site, we incubated FN-deficient (*Fn1*-KO) fibroblasts that express high levels of α5, αv, β1 and β3 integrins on their cell surface (*Figure 2—figure supplement 1A*) with blood plasma derived from either *Fn1^+/+^* or *Fn1^syn/syn^* mice. In line with our immunostaining of FN in tissues from *Fn1^syn/syn^* mice, *Fn1*-KO cells assembled fibrillar FN networks of indistinguishable complexity, fibril diameter and length with plasma from *Fn1^syn/syn^* and *Fn1^+/+^* mice, respectively (*Figure 2A*).

Next, we coated glass coverslips with plasma FN (pFN) purified from *Fn1^+/+^* or *Fn1^syn/syn^* mice (*Figure 2—figure supplement 1B–E*), seeded *Fn1*-KO fibroblasts and measured adhesion and spreading (*Figure 2B–E*). Adhesion of *Fn1*-KO cells to pFN^wt^ and pFN^syn^ began around 3 min after cell seeding and increased with time without noticeable differences (*Figure 2—figure supplement 1F*). While the formation of nascent adhesions (NAs) was similar on pFN^wt^ and pFN^syn^ (*Figure 2B*), the numbers as well as percentage of paxillin-positive focal adhesions (FAs) linked to stress fibers were significantly reduced in *Fn1*-KO fibroblasts seeded for 30 min on pFN^syn^ (*Figure 2D,E*) indicating that the transition from NAs to mature, stress fiber-anchored FAs is delayed on pFN^syn^. Furthermore, cell spreading determined as cell area at different time points after cell seeding onto pFN^syn^-coated substrates was also delayed in the first 30 min (*Figure 2C*). Time-lapse video microscopy confirmed the delayed cell spreading on pFN^syn^ and revealed unstable adhesions consisting of several cycles of binding and release from the substrate (see *Video 1*, *Video 2* and still images in *Figure 2—figure supplement 2*).

These findings indicate that the synergy site is dispensable for FN fibril formation but promotes the transition from NAs to FAs.

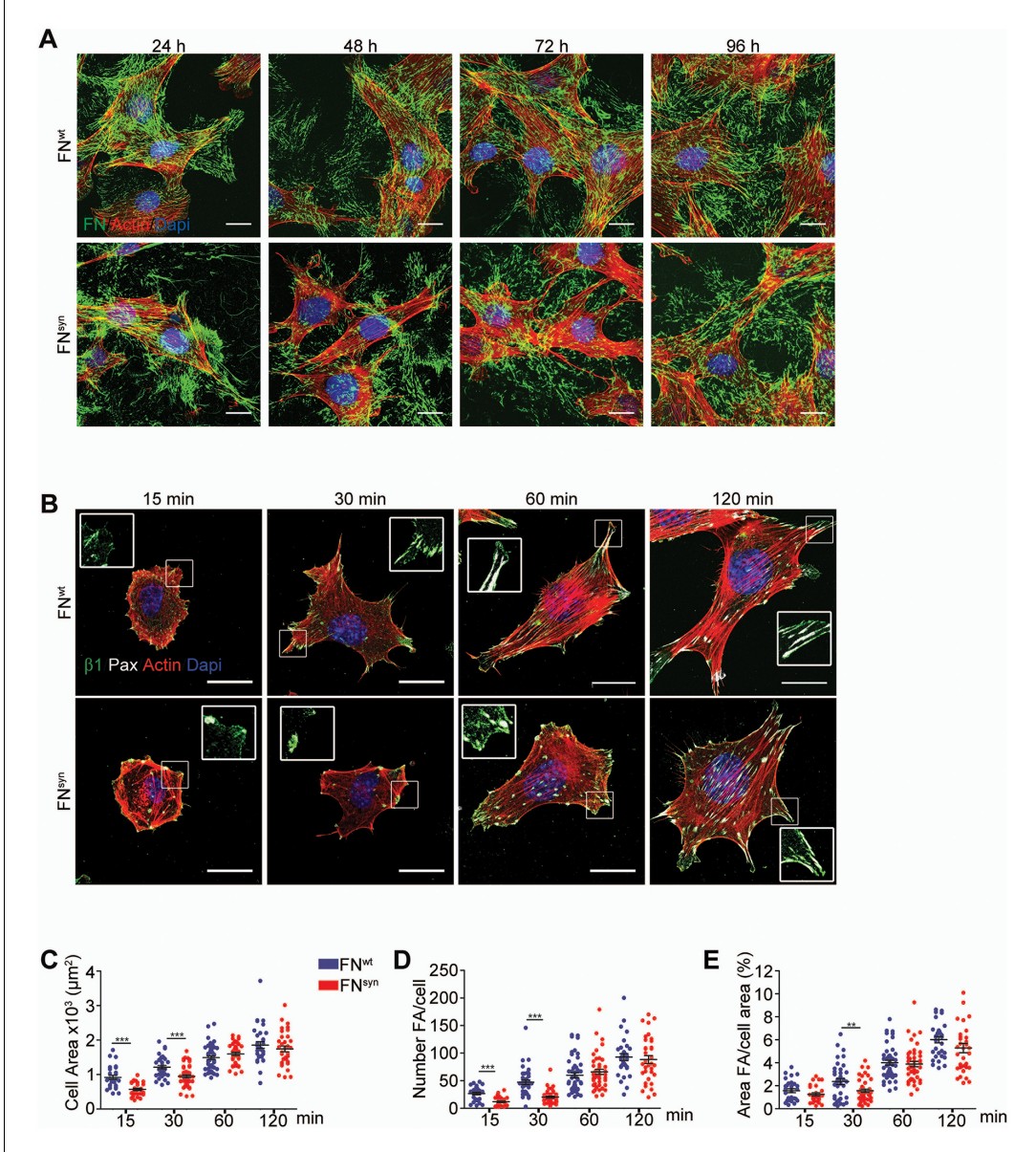

**Figure 2.** The FN synergy site is dispensable for FN fibrillogenesis, cell adhesion and spreading. (A) *Fn1*-Knock-Out (*Fn1*-KO) fibroblasts grown in 1% plasma derived from either *Fn1*$^{+/+}$ or *Fn1*$^{syn/syn}$ mice, fixed at the indicated times and stained for FN (green), F-actin stain (with Phalloidin; red) and nuclei (with DAPI; blue). Scale bar, 10 µm. (B) *Fn1*-KO cells seeded on pFN$^{wt}$ or pFN$^{syn}$, fixed at the indicated times and stained for F-actin (red), paxillin (white) and total $\beta$1 integrin (green). Scale bar, 20 µm. (C–E) Cell size (C), number of FAs per cell (D) and percentage coverage by FAs (paxillin-positive) (E) were quantified (n = 25 cells assessed from three independent experiments; mean ± sem). Statistical significances were calculated using the Student *t*-test; \*\*p<0.01 and \*\*\*p<0.001.

The following figure supplements are available for figure 2:

**Figure supplement 1.** Integrin surface levels and plasma FN purification and glass coating.

**Figure supplement 2.** Captures of life-time microscopy videos of *Fn1*-KO fibroblasts spreading on pFN$^{wt}$ or pFN$^{syn}$.

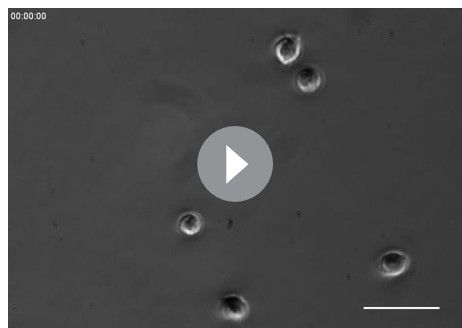

**Video 1.** Life-time microscopy video of *Fn1*KO fibroblasts on pFN^wt.

## The FN synergy site is required to tension FN-α5β1 bonds and to resist shear forces

It has been reported that HT1080 cells seeded on the FNIII7-10 polypeptide, increase adhesion strength to FN upon force application (*Friedland et al., 2009*). Therefore, we next tested whether the force-induced adhesion strengthening is FN-synergy site-dependent when *Fn1*-KO cells adhere to plasma-derived, purified full-length pFN^syn. We seeded overnight-starved *Fn1*-KO fibroblasts for 1 hr onto substrates coated with pFN^wt or pFN^syn and recombinant FNIII7-10^wt or FNIII7-10^syn polypeptides, respectively, and applied a hydrodynamic shear force with a spinning disk device (*García et al., 1998*). Typically, the number of *Fn1*-KO fibroblasts adhering to pFN^wt-coated coverslips and spun for 5 min decreased non-linearly with the applied force and followed a sigmoidal curve (*Figure 3—figure supplement 1*), whose inflection point (τ50) corresponds to the mean shear stress for 50% detachment, and hence to a quantitative measure of adhesion strength. Interestingly, the τ50 values of *Fn1*-KO cells decreased on purified full-length pFN^syn by 16% compared to pFN^wt (*Figure 3A*), and by 43% on FNIII7-10^syn fragment compared to FNIII7-10^wt, indicating that cells develop less adhesion strength on the synergy site-deficient pFN and that higher adhesion strengths arise on full-length FN compared to FNIII7-10 fragments.

Simultaneous engagement of the RGD motif and the synergy site was suggested to enable α5β1 and αIIbβ3 integrins to induce tensioned bonds, which form when receptor and ligand are in close proximity and hence, can be chemically cross-linked (*Shi and Boettiger, 2003*). To test the extent of bond tensioning on pFN^syn, we seeded (15, 30 and 60 min) serum-starved *Fn1*-KO fibroblasts onto pFN^wt- and pFN^syn-coated substrates, respectively, spun them and treated them with 3,3'-dithiobis (sulfosuccinimidyl propionate; DTSSP) to crosslink extracellular secondary amines that are within 1.2 nm proximity to each other. We found that the amount of α5 integrins crosslinked to FN in *Fn1*-KO fibroblasts was reduced to 60% on pFN^syn (*Figure 3B*). Upon spinning, *Fn1*-KO cells increased the proportion of α5 integrins crosslinked to pFN^wt. Importantly, in cells on pFN^syn, the tension was unable to increase the number of crosslinked bonds upon spinning and their numbers remained at the same levels as before spinning (*Figure 3B*), which altogether indicates that the spinning force strengthens α5β1-mediated adhesion to FN in a synergy site-dependent manner. Furthermore and in line with a report showing that the conversion of FN-α5β1 bonds from a relaxed to a tensioned state induces phosphorylation of focal adhesion kinase (FAK) on Y397 (*Guan et al., 1991*; *Kornberg et al., 1992*), pY397-FAK levels were reduced by 54% when cells were plated on pFN^syn compared to pFN^wt (*Figure 3C*). Importantly, phosphorylation of Y861-FAK, which occurs independent of substrate binding (*Shi and Boettiger, 2003*), was indistinguishable in cells seeded on pFN^wt or pFN^syn (*Figure 3C*). Since the intensity of FAK Y397 phosphorylation was shown to operate as a sensor for ECM rigidity (*Seong et al., 2013*), we conclude that fibroblasts attached to pFN^syn perceive insufficient information regarding substrate stiffness.

## αv-class integrins compensate for the absent FN synergy site

*Fn1*-KO cells express high levels of αv-class integrins (*Figure 2—figure supplement 1G*),

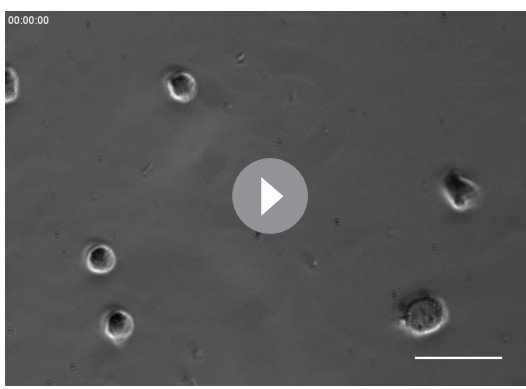

**Video 2.** Life-time microscopy video of *Fn1*KO fibroblasts on pFN^syn.

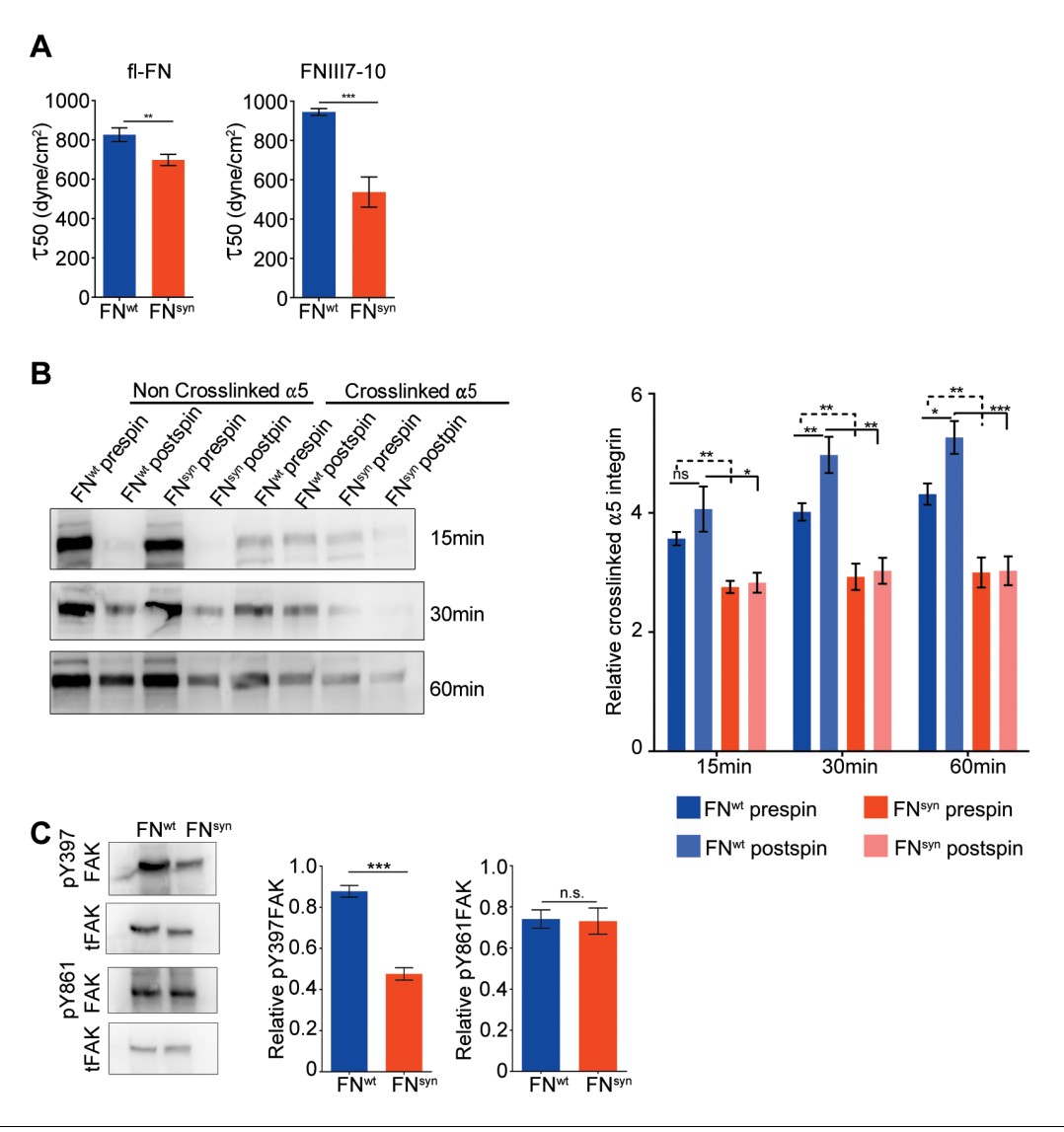

**Figure 3.** The FN synergy site is required to establish tensioned FN-α5β1 bonds. (**A**) Quantification of adhesion strength. $7 \times 10^5$ *Fn1*-KO cells attached onto purified, full-length (fl) pFN^wt or pFN^syn or FNIII7-10^wt or FNIII7-10^syn and spun with a spinning disk device (n = 7 independent experiments with fl-FN; n = 3 independent experiments with FNIII7-10; mean ± sem). (**B**) Western-blot analysis (left) and quantification (right) of cross-linked α5 integrins to pFN^wt or pFN^syn before and after applying shear forces (n = 6 independent experiments; mean ± sem). (**C**) Western-blot analysis (left) and quantification (right) of pY397- and pY861-FAK levels in *Fn1*-KO cells plated on pFN^wt or pFN^syn (n = 6 independent experiments; mean ± sem). Statistical significances were calculated using the Student *t*-test; *p<0.05, **p<0.01 and ***p<0.001.

The following figure supplement is available for figure 3:

**Figure supplement 1.** Representative spinning disk experiment showing the cell distribution profile against the shear force.

which could, at least in part, compensate for the absence of the synergy site during adhesion strengthening (*Figure 3*). To test this hypothesis, we seeded pan-integrin-null fibroblasts (pKO) reconstituted with β1-class integrins to express α5β1 (pKO-β1), or with αv integrins (pKO-αv) to express αvβ3 and αvβ5 integrins, or with both β1 and αv integrins (pKO-αv/β1) (*Schiller et al., 2013*) on pFN^wt- and pFN^syn-coated substrates and evaluated cell adhesion, spreading, and

adhesion site formation. From the three cell lines, only pKO-$\beta$1 cells exhibited reduced adhesion on pFN$^{syn}$ compared to pFN$^{wt}$ at all-time points analyzed (*Figure 4A*). Moreover, pKO-$\beta$1 cells had significantly fewer FAs, contained fewer stress fibers, and spread less on pFN$^{syn}$ compared to pFN$^{wt}$ (*Figure 4B–F*, see *Videos 3* and *4* and still images in *Figure 4—figure supplement 1*). Moreover, the areas of FAs determined with paxillin and $\beta$1 integrin stainings were significantly reduced on pFN$^{syn}$ compared to pFN$^{wt}$ (*Figure 4G,H*), which altogether suggests that pFN$^{syn}$-bound $\alpha$5$\beta$1 integrins fail to organize functional adhesion sites and to induce contractile stress fibers required for cell spreading. pKO-$\alpha$v cells adhered and spread similarly on pFN$^{wt}$ and pFN$^{syn}$, and developed comparably large, paxillin-positive FAs that were anchored to thick stress fibers (*Figure 4B,D*). Importantly, pKO-$\alpha$v/$\beta$1 cells also showed the same adhesion and spreading behavior, and developed similar adhesion sites on pFN$^{syn}$ indicating that $\alpha$v-containing integrins compensate for the absence of a functional synergy site (*Figure 4B,E*). Interestingly, the pKO-$\alpha$v/$\beta$1 cells do not show a delay in the transition from NAs to mature FAs on pFN$^{syn}$, as we observed with *Fn1*-KO cells, which could be due to the significantly higher $\beta$3 and lower $\alpha$5 integrin cell surface levels on pKO-$\alpha$v/$\beta$1 as compared to *Fn1*-KO cells (*Figure 2—figure supplement 1F*).

## The FN synergy site is dispensable for the on-rate of FN binding to $\alpha$5$\beta$1 integrins

Electron microscopy studies of the ligand-binding headpiece of integrin $\alpha$5$\beta$1 complexed with fragments of FN indicated no contact with the synergy site region while kinetic data suggested a role of the synergy site for enhancing the $K_{on}$ of the complex (*Takagi et al., 2003*). These findings gave rise to the hypothesis that the synergy site contributes to accelerate the initial encounter of $\alpha$5$\beta$1 with FN-RGD, which was in conflict with our observations that adhesion initiation was unaffected in *Fn1*-KO cells seeded on pFN$^{syn}$ (*Figure 2—figure supplement 1F*). To further test whether the synergy site is required for the FN binding on-rate, we quantified the probability of pKO-$\beta$1, pKO-$\alpha$v, pKO-$\alpha$v/$\beta$1 and pKO cells binding to FNIII7-10$^{wt}$ or FNIII7-10$^{syn}$ fragments and to purified full-length pFN$^{wt}$ or pFN$^{syn}$ using single-cell force spectroscopy (*Figure 4I,J*). To this end, a single cell was attached to the ConA-coated cantilever, lowered onto the FN with a speed of 1 µm/s until a contact force of 200 pN was recorded. After a very short contact time of $\approx$ 50 ms, cell and substrate were separated to detect the rupture of the few specific bonds formed between integrins and FN. On FNIII7-10$^{wt}$, the experiments revealed a 3-fold higher binding probability of pKO-$\alpha$v and pKO-$\alpha$v/$\beta$1 cells compared to pKO-$\beta$1 cells, indicating that $\alpha$v$\beta$3 integrins have a higher affinity for FN-RGD than $\alpha$5$\beta$1 integrins. Similar results were observed with full-length pFN$^{wt}$ or pFN$^{syn}$ (*Figure 4J*). Interestingly, however, full-length pFN showed higher binding probability than fragments for all cell lines tested including the pKO cells that lack integrin expression, which altogether suggests that in addition to integrins also other FN-binding cell surface receptor(s) contribute to the initial binding.

These findings indicate that the FN synergy site promotes the maturation of FAs but accelerates neither the rates of FN binding to $\alpha$5$\beta$1 integrins nor the formation of NAs.

## The FN synergy site compensates for $\alpha$IIb$\beta$3 integrin loss on platelets

To test whether the FN synergy site can also compensate for the loss of $\beta$3-class integrin expression in vivo, we generated homozygous compound mice carrying the *Fn1$^{syn}$* mutation and the *Itgb3* null mutation (*Itgb3$^{-/-}$*) (*Hodivala-Dilke et al., 1999*). *Itgb3*-null mice fail to express the widely expressed $\alpha$v$\beta$3 integrins and the platelet-specific $\alpha$IIb$\beta$3 integrin, and suffer from a bleeding disorder resembling human Glanzmann thrombasthenia. Around 87% of *Itgb3*-null mice are born and around 40% of them survive the first year of life (*Hodivala-Dilke et al., 1999*). To test how the *Fn1$^{syn}$* alleles affect development and survival of *Itgb3$^{-/-}$* mice, we intercrossed *Fn1$^{syn/+}$;Itgb3$^{+/-}$* as well as *Fn1$^{syn/syn}$;Itgb3$^{+/-}$* mice and obtained a total of 245 and 90 live offspring at P21, respectively (*Table 1* and *Table 1—source data 1*). Out of the 335 offspring altogether, one instead of the expected 38 compound homozygous *Fn1$^{syn/syn}$;Itgb3$^{-/-}$* mice survived to P21. The survivor died at the age of 5 months from excessive bleeding. To determine the time-point of lethality, embryos were collected at different gestation times and genotyped. While compound homozygous *Fn1$^{syn/syn}$;Itgb3$^{-/-}$* embryos were present at the expected Mendelian distribution until E15.5, no live embryos were present at E16.5 or later. Interestingly, mice with one wild-type *Itgb3* allele (*Fn1$^{syn/syn}$;Itgb3$^{-/+}$*) were normally

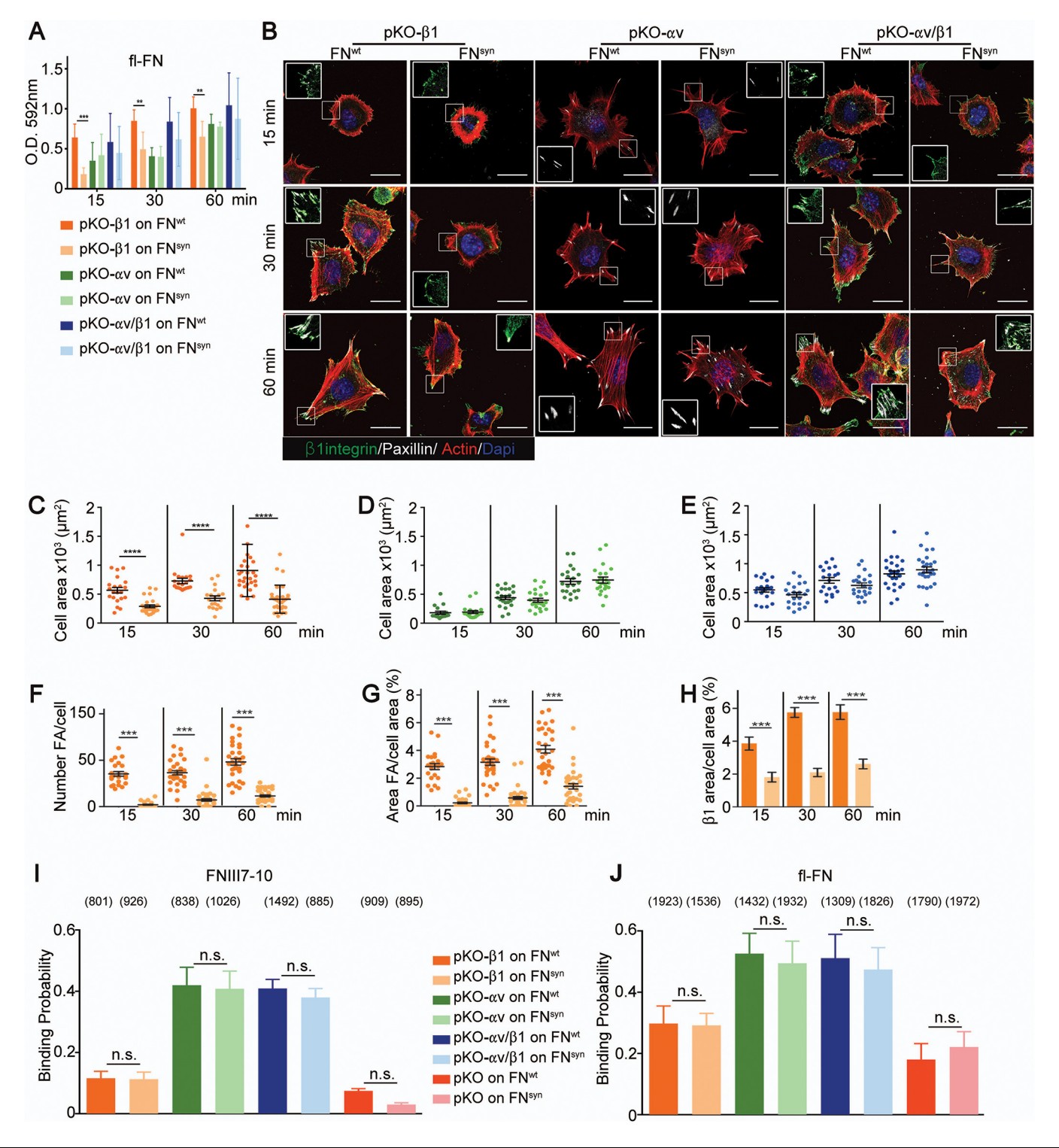

**Figure 4.** α5β1 integrins require the synergy site in FN to induce cell spreading. (**A**) Adhesion of pKO-β1, pKO-αv and pKO-αv/β1 fibroblasts seeded on pFN$^{wt}$ or pFN$^{syn}$ for indicated times (n = 3 independent experiments; mean ± sem). (**B**) pKO-β1, pKO-αv and pKO-αv/β1 fibroblasts were seeded on pFN$^{wt}$ or pFN$^{syn}$, fixed at the indicated times and stained for total β1 integrin (green), paxillin (white) and F-actin (red). Scale bar, 50 μm. (**C–E**) Quantification of cell area of pKO-β1 (**C**), pKO-αv (**D**) and pKO-αv/β1 (**E**) cells seeded on pFN$^{wt}$ or pFN$^{syn}$ for indicated times. (**F–H**) Quantification of the number of FAs (**F**), the percentage of FA coverage measured as paxillin-positive area (**G**) and the percentage of β1 integrin-positive areas referred to the total cell area (**H**) in pKO-β1 cells (n = 25 cells for each measurement and three independent experiments; mean ± sem). The binding probability

*Figure 4 continued on next page*

*Figure 4 continued*

of integrins to FNIII7-10$^{wt}$ or FNIII7-10$^{syn}$ fragments (I) and to full length (fl-FN) pFN$^{wt}$ or pFN$^{syn}$ (J) determined by single-cell force spectroscopy. Numbers in parentheses indicate events studied for each condition. Statistical significances were calculated using the Student *t*-test; *$p<0.05$, **$p<0.01$, ***$p<0.001$ and ****$p<0.0001$.

The following figure supplement is available for figure 4:

**Figure supplement 1.** Captures of life-time microscopy videos of pKO-$\beta$1 fibroblasts spreading on pFN$^{wt}$ or pFN$^{syn}$.

distributed, which altogether indicates that one $\beta$3 integrin allele is sufficient to compensate for normal development.

Compound homozygous *Fn1$^{syn/syn}$;Itgb3$^{-/-}$* embryos displayed multiple cutaneous hemorrhages and edema, which were first visible at E11.5/12.5 (*Figure 5A* and *Figure 5—figure supplement 1A*) and then spread over the whole body at E15.5 (*Figure 5B*). Interestingly, at E12.5 the bleeds were visible at sites where the lymphatic vessels form (arrowheads in *Figure 5A*) and therefore, we hypothesized that the newly formed lymphatic vessels fail to separate from the cardinal vein, which occurs between E11-13 (*Carramolino et al., 2010*). In line with our hypothesis, Lyve1-positive lymphatic vessels in the skin of *Fn1$^{syn/syn}$;Itgb3$^{-/-}$* embryos were dilated and covered with ectopic $\alpha$-smooth muscle actin ($\alpha$-SMA)-positive cells and filled with Ter119-positive erythroblasts (*Figure 5C–E*). In contrast, lymphatic vessels in the skin of *Itgb3*-null or wild-type littermates neither contained Ter119-positive cells nor were surrounded with $\alpha$-smooth muscle actin-positive cells.

The separation of the primary lymphatic sac from the cardinal vein is driven by platelet adhesion to and aggregation at the lymphatic endothelium (*Carramolino et al., 2010*; *Uhrin et al., 2010*). We therefore hypothesized that the platelet functions are severely compromised in *Fn1$^{syn/syn}$;Itgb3$^{-/-}$* embryos as they lack $\alpha$IIb$\beta$3-mediated binding to fibrinogen and FN (*Figure 6A*), as well as the ability to strengthen adhesion and signaling via $\alpha$5$\beta$1 integrin-mediated binding to FN. To test the hypothesis, we performed spreading assays as well as adhesion assays under flow with wild-type or *Itgb3$^{-/-}$* platelets. The mean spreading area of wild-type platelets seeded for 60 min on fibrinogen, pFN$^{wt}$, and pFN$^{syn}$ was 15–16 $\mu$m$^2$. As expected, *Itgb3$^{-/-}$* platelets failed to spread on fibrinogen (mean spreading area of 4.6 $\mu$m$^2$). Furthermore, they showed a reduced mean spreading area of 8.9 $\mu$m$^2$ on pFN$^{wt}$ and failed to spread on pFN$^{syn}$ (mean spreading area of 3.8 $\mu$m$^2$) (*Figure 6B,C*). Application of shear flow reduced adhesion of wild-type platelets to pFN$^{syn}$ by 10-fold compared to pFN$^{wt}$, while adhesion of *Itgb3*-null platelets was lost on fibrinogen as well as pFN$^{syn}$, and only slightly diminished on pFN$^{wt}$ (*Figure 6D,E*). Importantly, adhesion and spreading of platelets isolated from *Itgb3$^{-/-}$* mice to collagen were unaffected, irrespective of whether shear flow was applied or not (*Figure 6C,E*).

These in vitro experiments demonstrate that adhesion of $\alpha$IIb$\beta$3-deficient platelets to wild-type FN is partially compensated by $\alpha$5$\beta$1 integrins in a FN synergy site-dependent manner, and that $\alpha$5$\beta$1 as well as $\alpha$IIb$\beta$3 integrins require the FN synergy site for stabilizing platelet adhesion to FN, under shear flow.

## The FN synergy site compensates for $\alpha$v$\beta$3 during vessel maturation

The absence of $\alpha$5$\beta$1 integrins leads to vascular defects (*Abraham et al., 2008*). To test whether vascular abnormalities due to an impaired $\alpha$5$\beta$1 function contribute to the severe bleeds and the lethality of *Fn1$^{syn/syn}$;Itgb3$^{-/-}$* embryos, we analyzed the mural coverage and anchorage to the ECM. While immunostaining of E11.5 whole mount embryos with an anti-PECAM-1 antibody revealed that the vessels in the trunk of *Fn1$^{syn/syn}$;Itgb3$^{-/-}$* embryos showed normal sprouting

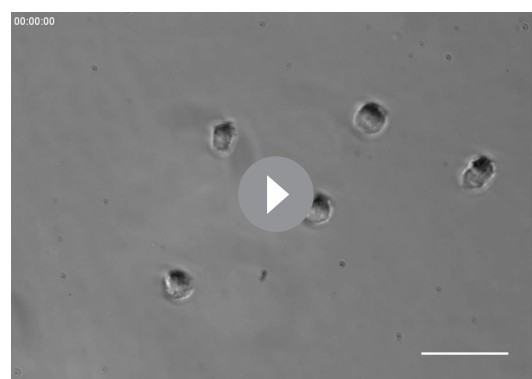

**Video 3.** Life-time microscopy video of pKO-$\beta$1 fibroblasts on pFN$^{wt}$.

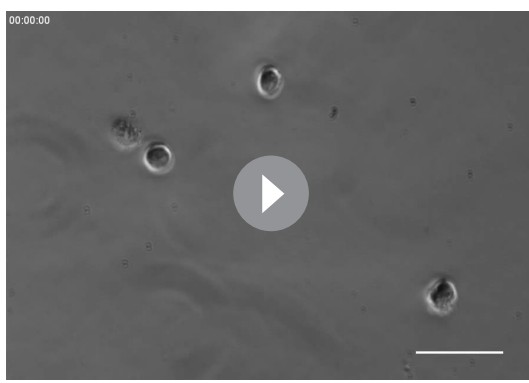

**Video 4.** Life-time microscopy video of pKO-$\beta$1 fibroblasts on pFN$^{syn}$.

(*Figure 5—figure supplement 1B*), the arteries and veins of the dermal vasculature of E15.5 embryos were tortuous and irregularly covered with $\alpha$-SMA-positive cells (*Figure 7A*). Furthermore, the vascular network was less intricate and had significantly fewer branching points in *Fn1$^{syn/syn}$;Itgb3$^{-/-}$* embryos compared to wild-type littermates (*Figure 7B*). Interestingly, collagen IV immunostaining indicated that many small vessels in E15.5 *Fn1$^{syn/syn}$;Itgb3$^{-/-}$* embryos lacked a clear lumen and PECAM-1 immunosignals (see arrowheads in *Figure 7C*). They probably represent retracted vessels and were significantly more frequent in *Fn1$^{syn/syn}$;Itgb3$^{-/-}$* embryos compared to wild-type, *Fn1$^{syn/syn}$; Itgb3$^{+/+}$* and *Fn1$^{+/+}$;Itgb3$^{-/-}$* littermates (*Figure 7D*). Moreover, small vessels in *Fn1$^{syn/syn}$;Itgb3$^{-/-}$* embryos were often less covered by pericytes. Instead, NG2-positive pericytes were either detached or formed patchy aggregates on the vessel surface (see arrowheads in *Figure 7E*). Altogether, these observations indicate that the vessel wall coverage and stability are decreased in the *Fn1$^{syn/syn}$;Itgb3$^{-/-}$* embryos and probably contribute to their severe hemorrhages.

## Discussion

Although cell-based studies suggested that the FN synergy site is required for $\alpha$IIb$\beta$3 and $\alpha$5$\beta$1 integrin function, the in vivo evidence was missing and the mechanistic property controversial. We report here the characterization of a mouse strain, in which the synergy site of FN (*Fn1$^{syn}$*) was disrupted. Contrary to expectations, the *Fn1$^{syn/syn}$* mice were born without developmental defects indicating that the synergy site is dispensable for organogenesis and tissue homeostasis. However, when *Fn1$^{syn/syn}$* mice are exposed to stress such as tail bleeding and arteriole injury, or the genetic ablation of the FN-binding $\beta$3-class integrins ($\alpha$v$\beta$3, $\alpha$IIb$\beta$3), the synergy site becomes essential for cells that have to resist or produce high forces such as platelets and vascular smooth muscle cells (*Figure 8*).

**Table 1.** Progeny of *Fn1$^{syn/+}$;Itgb3$^{+/-}$* x *Fn1$^{syn/+}$;Itgb3$^{+/-}$* intercrosses.

| Age | Num. | *Fn1$^{syn/syn}$ Itgb3$^{+/-}$* | *Fn1$^{syn/syn}$ Itgb3$^{+/+}$* | *Fn1$^{syn/syn}$ Itgb3$^{-/-}$* | *Fn1$^{+/syn}$ Itgb3$^{+/-}$* | *Fn1$^{+/syn}$ Itgb3$^{+/+}$* | *Fn1$^{+/syn}$ Itgb3$^{-/-}$* | *Fn1$^{+/+}$ Itgb3$^{+/-}$* | *Fn1$^{+/+}$ Itgb3$^{+/+}$* | *Fn1$^{+/+}$ Itgb3$^{-/-}$* |
|---|---|---|---|---|---|---|---|---|---|---|
| E11.5 | 36 | 6 (16.7%) | 1 (2.8%) | 1 (2.8%) | 10 (27.8%) | 6 (16.7%) | 4 (11.1%) | 5 (13.9%) | 2 (5.6%) | 1 (2.8%) |
| E14.5 | 23 | 2 (8.7%) | 2 (8.7%) | 2 (8.7%) | 5 (21.7%) | 5 (21.7%) | 1 (4.3%) | 1 (4.3%) | 4 (17.3%) | 1 (4.3%) |
| E15.5 | 121 | 12 (9.9%) | 5 (4.1%) | 3 (2.5%) | 39 (32.2%) | 5 (15.4%) | 12 (9.9%) | 14 (11.6%) | 17 (14%) | 4 (3.3%) |
| E16.5 | 16 | 2 (12.5%) | 1 (6.25%) | 0 | 5 (31.5%) | 1 (6.25%) | 1 (6.25%) | 3 (37.5%) | 1 (6.25%) | 2 (12.5%) |
| E17.5 | 16 | 2 (12.5%) | 0 | 0 | 6 (23%) | 3 (19%) | 2 (12.5%) | 2 (12.5%) | 1 (8%) | 0 |
| P 21 | 245 | 33 (13.5%) | 32 (13%) | 0 | 57 (23%) | 46 (18.7%) | 13 (5.3%) | 35 (14.4%) | 17 (3.9%) | 12 (4.9%) |
| Mendelian Distribution | 100 | 12.5% | 6.25% | 6.25% | 25% | 12.5% | 12.5% | 12.5% | 6.25% | 6.25% |

**Source data 1.** Progeny of *Fn1$^{syn/syn}$;Itgb3$^{+/-}$* x *Fn1$^{syn/syn}$;Itgb3$^{+/-}$* crosses

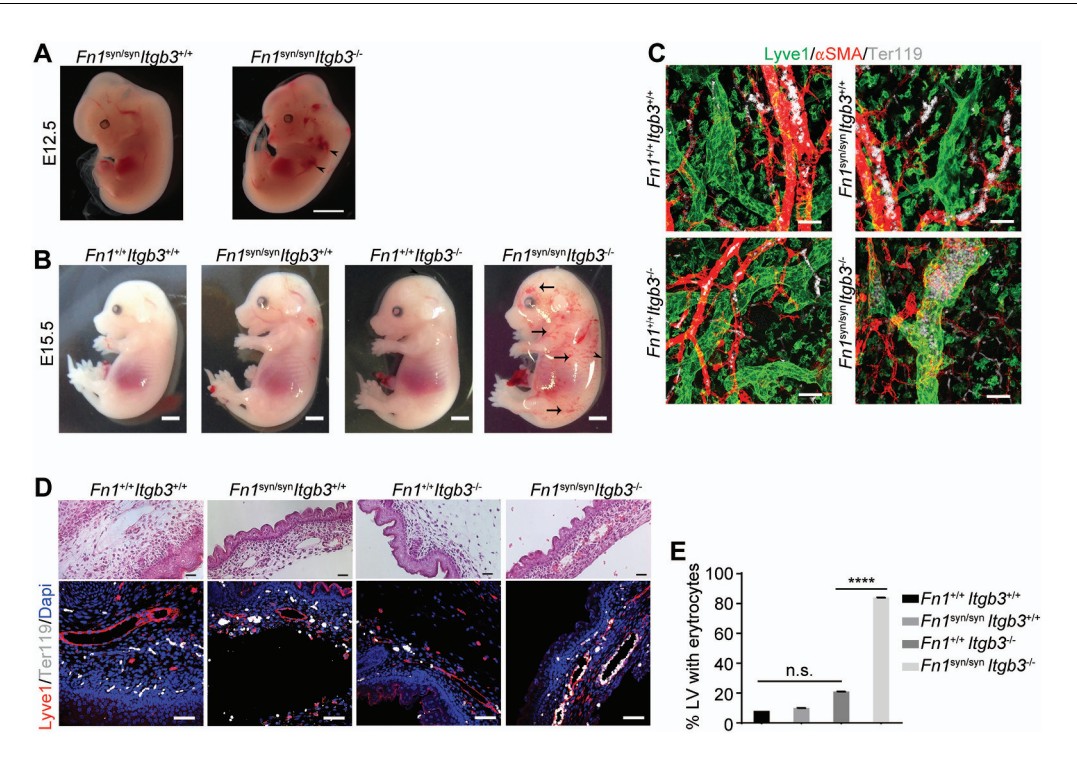

**Figure 5.** *Fn1^syn/syn^;Itgb3^-/-^* mice suffer from severe hemorrhages and fail to separate the blood and lymphatic vasculatures. (**A**) E12.5 *Fn1^syn/syn^;Itgb3^-/-^* embryos display hemorrhages in the jugular and axilar areas in the left side (arrowheads). Scale bar, 50 mm. (**B**) Representative images from E15.5 littermates embryos resulting from *Fn1^syn/+^;Itgb3^-/+^* intercrosses. Compound *Fn1^syn/syn^;Itgb3^-/-^* embryos display cutaneous edema (arrowhead) and abundant skin hemorrhages (arrows); scale bars, 50 mm. (**C**) Skin whole-mount from E15.5 embryos showing Lyve1-positive lymphatic vessels (green), αSMA-positive blood vessels (red) and Terr119-positive erythrocytes (white). The lymphatic vessels of compound *Fn1^syn/syn^;Itgb3^-/-^* embryos are dilated, covered by ectopic αSMA-positive cells and filled with erythrocytes. Scale bar, 50 μm. (**D**) Representative images of skin sections stained with H and E (upper panel) and Lyve1 and Terr119 (lower panel) showing erythrocytes in lymphatic vessels. Scale bar, 50 μm. (**E**) Quantification of the percentage of lymphatic vessels filled with Ter119-positive erythrocytes (n = 40 vessels counted per embryo, in two embryos per each genotype; mean ± sem). Statistical significances were calculated using the Student *t*-test: ****p<0.0001.

The following figure supplement is available for figure 5:

**Figure supplement 1.** Blood vessel formation in *Fn1^syn/syn^; Itgb3^-/-^* embryos.

Ablation of the *Fn1* gene in mice, as well as the simultaneous ablations of the *Itga5/Itgav* integrin genes in mice arrests development at embryonic day 8.5 (E8.5) due to defects in the formation of mesoderm and mesoderm-derived structures (*George et al., 1993*; *Georges-Labouesse et al., 1996*; *Yang et al., 1999*). The replacement of the FNIII10 RGD motif with the RGE in mice also affects mesoderm development, although less severe and restricted to the vascular system and to the posterior region of the developing embryo (*Takahashi et al., 2007*; *Girós et al., 2011*). Interestingly, these defects resemble those observed in *Itga5*-deficient mice indicating that the RGE mutation is sufficient to abrogate α5β1 integrin function and that the synergy site cannot compensate for a dysfunctional RGD motif. Furthermore, the normal development of *Fn1^syn/syn^* mice also excludes an essential role of the synergy site for α5 integrin function in vivo (*Grant et al., 1997*; *Krammer et al., 2002*). A reduced α5β1 integrin function would probably have occurred if the synergy site would indeed guide the binding pocket of α5β1 towards the RGD motif and increase the FN-binding on-rate (*Takagi et al., 2003*). However, the absence of obvious 'α5β1-loss-of-function defects' (*Yang et al., 1993*) in *Fn1^syn/syn^* mice and the normal FN-binding on-rates of pKO-β1 cells in single-cell force spectroscopy experiments indicate that the synergy site is probably dispensable to accelerate α5β1 integrin-FN binding.

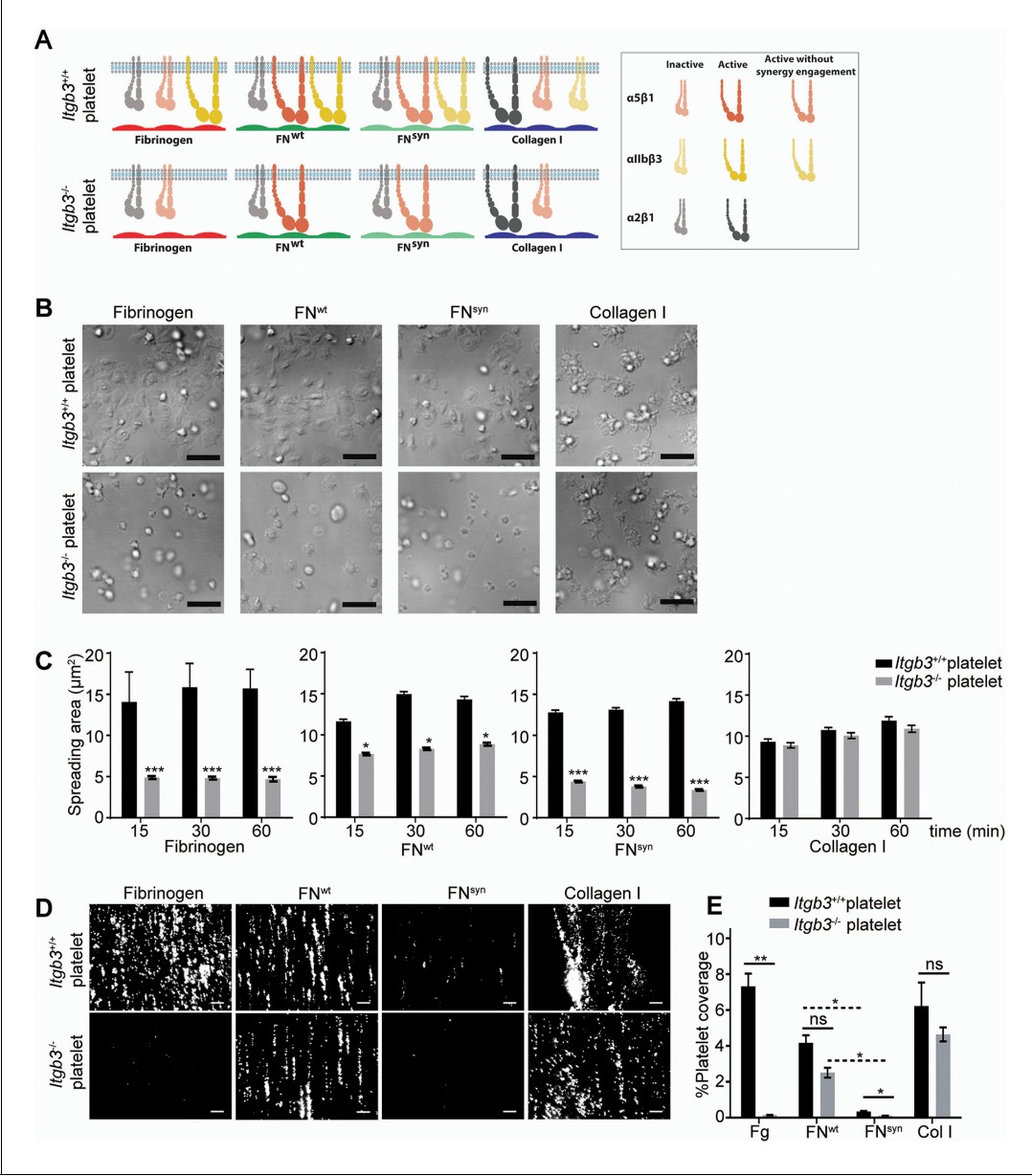

**Figure 6.** Shear flow exposed platelets fail to adhere to pFN^syn. (**A**) Cartoon showing the platelet integrins that can be ligated to the different substrates used in the experiments. The color intensity of the integrin denotes whether the integrin is active or inactive. (**B**) Spreading of *Itgb3*^+/+^ and *Itgb3*^-/-^ platelets after 1 hr on fibrinogen, pFN^wt^, pFN^syn^ and type I collagen. Scale bars, 10 μm. (**C**) Quantification of the platelet area at indicated times (n = 100 platelets per each condition in three independent experiments; mean ± sem). (**D**) Representative figures of fluorescently labeled *Itgb3*^+/+^ or *Itgb3*^-/-^ platelets seeded on indicated substrates and exposed to shear flow. Scale bar, 40 μm. (**E**) Platelet coverage after 10 min shear flow of $1000 \ s^{-1}$. (n = 10 pictures per experiment, four independent experiments for each condition; mean ± sem). Statistical significances were calculated using the Student *t*-test; *p<0.05, **p<0.01 and ***p<0.001.

We also demonstrate an unexpected, compensatory role between the FN synergy site and αvβ3 integrins for the vascular coverage by smooth muscle cells. Apparently, the high myosin II-induced forces generated by these cells are only efficiently absorbed with either high αv-class integrin surface levels or a fully functional FN. Whether a similar functional relationship operates also during paraxial mesoderm, whose formation critically depends on the expression of α5β1 and αv-class integrins (*Yang et al., 1999*), cannot be deduced from our experiments. However, the normal mesoderm formation in *Fn1*^syn/syn^ mice indicates that mesodermal cells require α5β1 to bind the RGD motif but

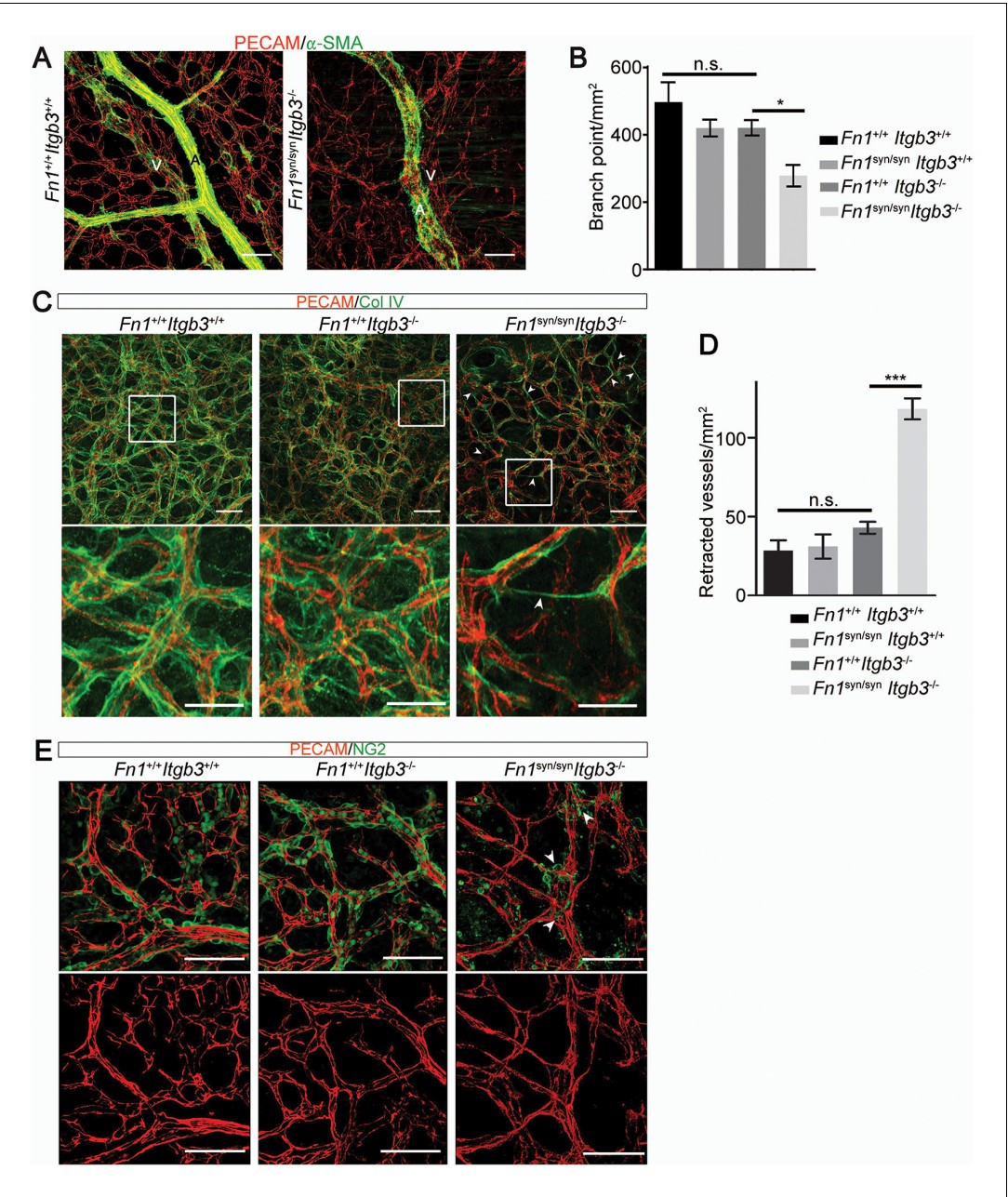

**Figure 7.** Malformed blood vessels in *Fn1*<sup>syn/syn</sup>;*Itgb3*<sup>-/-</sup> embryos. (**A**) PECAM-positive endothelial cells (red) and α-SMA-positive smooth muscle cells (green) in dermal whole mounts from E15.5 *Fn1*<sup>+/+</sup>;*Itgb3*<sup>+/+</sup> and *Fn1*<sup>syn/syn</sup>; *Itgb3*<sup>-/-</sup> littermate embryos indicate veins (V) and arteries (A). (**B**) Quantification of the number of branching points (n = 10–15 images of 2–3 embryos; mean ± sem). (**C**) Vascular basement membranes in dermal whole mounts from E15.5 *Fn1*<sup>+/+</sup>;*Itgb3*<sup>+/+</sup>, *Fn1*<sup>+/+</sup>;*Itgb3*<sup>-/-</sup> and *Fn1*<sup>syn/syn</sup>;*Itgb3*<sup>-/-</sup> littermate embryos stained for type IV collagen (green) and PECAM-positive endothelial cells (red). Arrowheads show small vessels lacking lumen. (**D**) Quantification of retracted vessels (n = 14–23 from 4–7 embryos; mean ± sem). (**E**) PECAM-positive endothelial cells (red) and NG2-positive pericytes (green) in dermal whole-mounts from E15.5 *Fn1*<sup>+/+</sup>;*Itgb3*<sup>+/+</sup>, *Fn1*<sup>+/+</sup>;*Itgb3*<sup>-/-</sup> and *Fn1*<sup>syn/syn</sup>;*Itgb3*<sup>-/-</sup> littermate embryos. Note pericytes are sparse, absent or aggregate on mutant vessels (arrowheads). Statistical significances were calculated using the Student *t*-test; *p<0.05, and ***p<0.001. Scale bars, 50 μm.

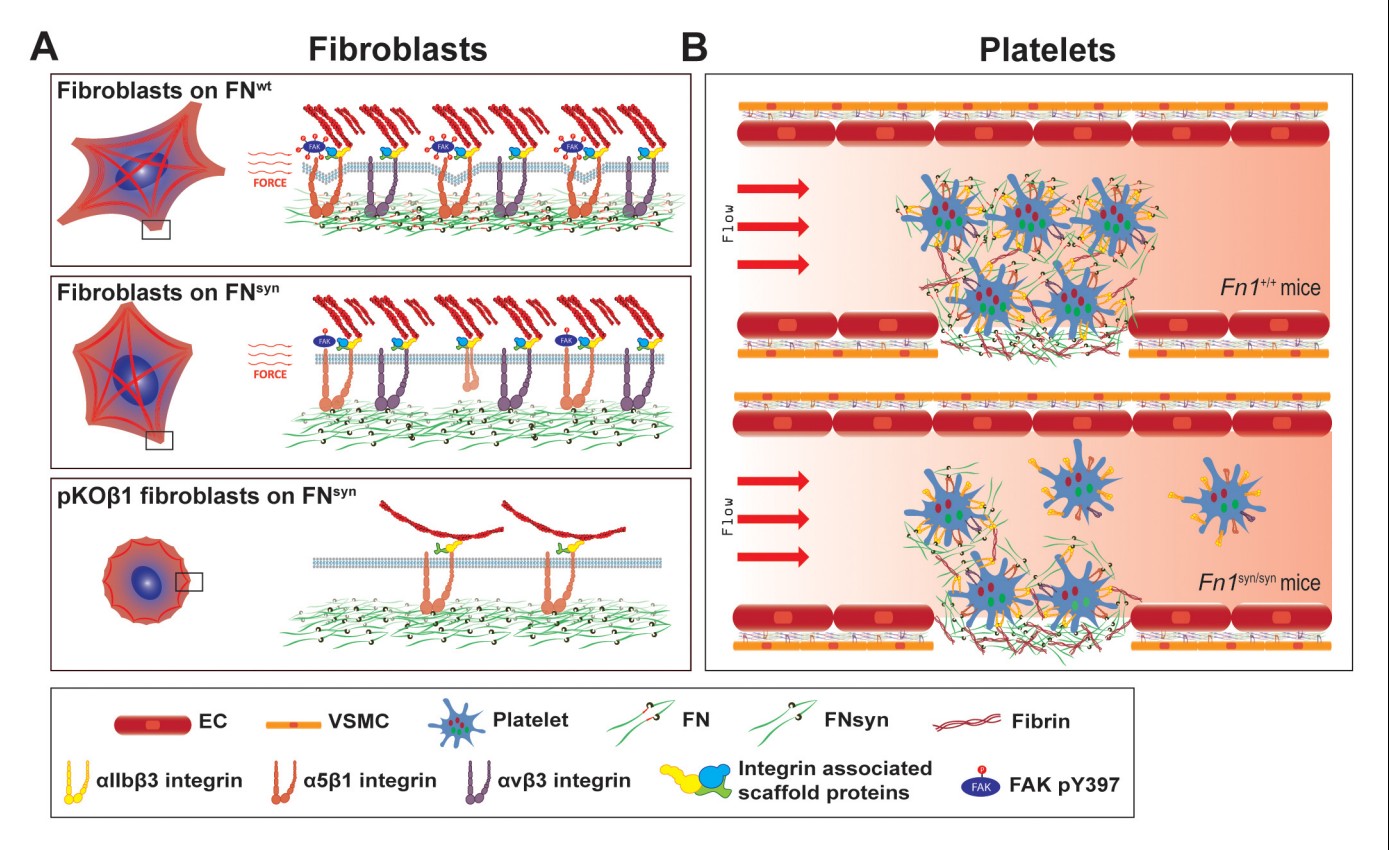

**Figure 8.** The major role of the FN synergy site is to re-enforce cell adhesion. (**A**) Hydrodynamic shear force-exposed fibroblasts seeded on a FN^{wt}-coated surface form catch-bonds that strengthen α5β1 integrin-mediated adhesions to FN and trigger phosphorylation of Y397-FAK (upper image). On FN^{syn}-coated surfaces, the αvβ3 integrins compensate for the absent synergy site allowing fibroblast adhesion and the reduced α5β1 binding strength leads to diminished phosphorylation of pY397-FAK (middle image). The elimination of αv-class integrins decreases cell adhesion on FN^{syn}-coated surfaces, reduces cell spreading and delays the maturation of FA and fibrillar adhesions (lower image). (**B**) Platelets in *Fn1^{+/+}* mice form tight aggregates on injured vessel walls that withstand the shear forces of the blood flow (upper image), while platelets in an injured vessel in *Fn1^{syn/syn}* mice fail to withstand the blood flow leading to a delayed thrombus formation (lower image). Endothelial cells (EC); vascular smooth muscle cells (VSMC).

not the synergy site. Interestingly, we also find clear evidence for a compensatory role of the FN synergy site and αv-class integrins for cell spreading and FA maturation in vitro (*Figure 8*), which differs from previous reports showing that α5β1 and αv-class integrins have non-overlapping functions for inducing myosin contractility (*Schiller et al., 2013*) or controlling directional migration (*Missirlis et al., 2016*).

It is well known that platelet adhesion to pFN, mediated by αIIbβ3 with contributions from α5β1 and other integrins, plays a critical role for hemostasis (*Wang et al., 2014*; *Ni et al., 2003b*). Our findings revealed that the FN synergy site is critically important for the adhesion of wild type platelets in in vitro flow chamber settings, while impaired spreading or defects under static adhesion only arise when αIIbβ3 expression is lost. These findings indicate that the synergy site plays a central role as soon as force is applied to the bonds between FN and platelet integrins. A similar force-dependent requirement of the synergy site was also apparent after arteriolar injuries in vivo, which showed that *Fn1^{syn/syn}* mice display diminished platelet adhesion and delayed thrombus formation. This requirement of the synergy site for platelets to resist the shear forces of the blood flow also provides a rational explanation for the prolonged bleeding times observed in *Fn1^{syn/syn}* mice after tail tip excisions. Interestingly, the in vivo platelet dysfunctions in *Fn1^{syn/syn}* mice as well as *Itgb3^{-/-}* mice profoundly aggravate in compound *Fn1^{syn/syn};Itgb3^{-/-}* embryos where they cause fatal bleedings and an insufficient platelet-mediated separation of the lymphatic vessels from the cardinal vein. Since murine platelets contain around 100 times more αIIbβ3 than α5β1 integrins (*Zeiler et al., 2014*) the

compensation of the entire αIIbβ3 pool by the minor α5β1-FN$^{wt}$ complexes underscores the fundamental role of the adhesion strengthening property of the synergy site in FN.

Our mouse strain will allow now to test how the synergy site-mediated adhesion re-enforcement affects the course of tissue fibrosis or cancer development and progression, which are heavily influenced by integrin surface levels as well as tissue rigidity that in turn is modulated by the strength of integrin-ligand bonds (*Zilberberg et al., 2012*; *Laklai et al., 2016*).

## Materials and methods

### Animals

Mice were housed in special pathogen free animal facilities. All mouse work was performed in accordance with the Government of the Valencian Community (Spain) guidelines (permission reference A1327395471346) and with the Government of Upper Bavaria. Mice containing the integrin β3 deletion were bred under the permission reference 55.2-1-54-2532-96-2015. The tail-bleeding and cremaster muscle venules injury assays performed under the permission reference 55.2-1-54-2532-115-12.

### Generation of *Fn1$^{syn/syn}$* knockin mice

A 129/Sv mouse PAC clone was used to construct the targeting vector (*Figure 1—figure supplement 1A*), which consisted of a 2.1 kb fragment containing exons 26 and 27, a neomycin cassette flanked by *loxP* sites, a 2.4 kb fragment containing the exon 28 carrying the mutated nucleotides, and a 3.5 kb fragment with the exons 29 to 32. The targeting construct was linearized with NotI and electroporated into R1 embryonic stem (ES) cells. Approximately 300 G418-resistant clones were isolated and screened by Southern blot for homologous recombination. The genomic DNAs were digested with SacI, XbaI or BstEII and probed with external probes 1 and 2 (*Figure 1—figure supplement 1A*). Two correctly targeted clones were injected into C57BL/6 host blastocysts to generate germline chimeras. The *Fn1$^{syn-neo/+}$* mice were crossed with a deleter-Cre strain to eliminate the *loxP* flanked neomycin cassette. The elimination of neomycin was analyzed by Southern blot, genomic DNA was digested with Eco RI and probed with probe 3 (*Figure 1—figure supplement 1B*). The *Fn1$^{syn/+}$* mice were intercrossed to generate homozygous *Fn1$^{syn/syn}$* mice. The following primers were used to genotype the mouse strain by PCR: 5'-TCACAAGGAAACCAGGGAAC-3' (forward); 5'-CCGTTTTCACTCTCGTCAT-3' (reverse).

### Cell lines

The mouse *Fn1*-KO cell line and the integrin pan-KnockOut fibroblast lines were isolated from a mouse kidney and immortalized by retroviral delivery of the SV40 large T. To generate *Fn1*-KO cells, the *Fn1* gene was deleted from *Fn1$^{flox/flox}$* with the adenoviral transduction of the *Cre* recombinase. Integrin pKO fibroblasts were generated as described by *Schiller et al. (2013)*. *Itgav*$^{flox/flox}$; *Itgb1*$^{flox/flox}$; *Itgb2*$^{-/-}$;*Itgb7*$^{-/-}$ immortalized fibroblasts were treated with adenoviral *Cre* recombinase and reconstituted with mouse *Itgb1* and/or *Itgav* integrin cDNAs to generate pKO-β1, pKO-αv and pKO-αv/β1 cells. The cells were provided by H. Schiller (Max-Planck Institute for Biochemistry, Martinsried, Germany). Cell lines were not tested for mycoplasma.

### Antibodies

For flow cytometry, we used the following antibodies conjugated to PE: 1:200; hamster anti-β1 integrin (102207, from from BioLegend, San Diego, CA, USA), rat anti-α5 integrin (557447, from from PharMingen, Madrid, Spain), hamster anti-β3 integrin (12–0611, from BD Bioscience, Madrid, Spain) and rat anti-αv integrin (551187, from from PharMingen). For immunostainings or western blots, we used the following antibodies: rabbit anti-β1 integrin (obtained from Reinhard Fässler) IF 1:500; rabbit anti-α5 integrin (4705, from Cell Signaling, Barcelona, Spain) WB 1:2000; rabbit anti-pTyr397FAK (44–624G, from Biosource, Madrid, Spain) WB 1:1000; rabbit anti-pTyr861FAK (44–626G, from Biosource) WB 1:1000; rabbit anti-FAK (06–543, from Millipore, Ille de France, France) WB 1:1000; rabbit anti-Fibronectin (AB2033, from Millipore) WB 1:2000, IF 1:300–500; mouse anti-Paxillin (610051, from PharMingen) IF 1:300; rabbit anti-Lyve1 (ab14914, from Abcam, Cambridge, UK) IF 1:300, WM 1:100; rat anti-Ter119 (09082D, from PharMingen) IF 1:400,

WM 1:100; rat anti-PECAM (553370, from PharMingen) IF 1:300, WM 1:100; rabbit anti-Collagen IV (2150–1470, from Bio-Rad, Madrid, Spain) WM 1:100; mouse anti-smooth muscle actin conjugated with Cy3 (A2547, from Sigma-Aldrich, Madrid, Spain) IF 1:500, WM 1:200; anti-NG2 chondroitin sulfate proteoglycan (AB5320, from Merck Millipore) WM 1:100; mouse anti-gamma chain fibrinogen (ab119948, from Abcam), WB 1:200; rabbit anti-laminin-1 (L9393, from Sigma-Aldrich) WM 1:100. F-actin was stained with Phalloidin coupled with TRITC (P1951, from Sigma-Aldrich) IF 1:500. For immunofluorescence the following secondary antibodies were used diluted 1:400: goat anti-rabbit conjugated with Alexa488 (A11008); goat anti-rabbit conjugated with A546 (A11010); donkey anti-mouse conjugated with A647 (A31571); goat anti-rat conjugated with A488 (A21208); goat anti-rat conjugated with A546 (A11081); goat anti-rat conjugated with A647 (A21247) (all from Invitrogen, Madrid, Spain). For western-blot, goat anti–rabbit conjugated with horseradish peroxidase (172–1019, from Bio-Rad) was used as secondary antibody.

## Purification of plasma fibronectin (pFN)

Blood was collected from $Fn1^{+/+}$ and $Fn1^{syn/syn}$ mice using 0.5 M EDTA as anticoagulant in non-heparinized capillaries, centrifuged at 3000 rpm for 20 min and the pFN was purified from the supernatant (plasma) using Gelatin-Sepharose (GE Healthcare Life Sciences, Valencia, Spain) affinity chromatography (Retta et al., 1999) adapted to minicolumns (Poly-Prep, Bio-Rad). Briefly, the columns were washed with 0.5 NaCl in 10 mM Tris-HCl pH7.4 and pFN was eluted with 2 M urea in TBS (0.15 M NaCl in 10 mM Tris-HCl, pH 7.4) and dialyzed against TBS. Purified FN was analyzed by 8% SDS-PAGE and stained with Coomassie brilliant blue, and by Western blot.

## Production of the FNIII7-10 fragment

We used the human cDNA encoding the FNIII7-10 fragment and subcloned in the expression vector pET-15b (Takahashi et al., 2007). To generate the FNIII7-10$^{syn}$ we mutated by site-directed mutagenesis the two arginines in the synergy sequence: DRVPHSRN>DAVPHSAN. We performed two rounds of PCR using the following primers: 5´-GATGCGGTGCCCCACTCTCGGAAT-3´ (forward) and 5´-GATGCGGTGCCCCACTCTGCGAAT-3´ (forward) and the complementary reverse primers. The expression of recombinant FN fragments was done in the E. coli strain Rosetta T1R. Purification was performed using TALON Metal Affinity chromatography (Clontech, Saint Germain en Laye, France). Finally the protein was obtained by gel filtration chromatography using Superdex 200 10/300 GL columns (GE Healthcare) and Superdex Size Exclusion Media (GE Healthcare, Valencia, Spain) and eluted in PBS.

## Adsorption of purified pFN onto glass

Glass coverslips of 18 mm diameter were poly-maleic anhydride-1-octadecene (POMA; Polysciences Inc)-treated (Prewitz et al., 2013) and coated with 0.1–10 µg/ml of purified mouse pFN during 1 hr at RT, followed by a blocking step of 1 hr using 1% BSA in PBS. To quantify the adsorbed FN, the coverslips were then incubated for 2 hr at RT with anti-FN antibodies (Ab; diluted 1/300 in blocking solution), washed, incubated with anti-rabbit Ab conjugated-HRP (diluted 1/500 in blocking solution) 1 hr at RT and finally treated with 50 µl of 2,2'-azino-bis(3-ethylbenzthiazoline-6-sulfonic acid (ABTS; Peroxidase substrate kit, Vector SK-4500) for 30 min in the dark. The ABTS-containing solution was collected and the absorbance was measured at 405 nm.

## Cell adhesion assay

96 well plates were coated with 10 µg/ml of pFN or poly-lysine (Sigma-Aldrich, Madrid, Spain) or 3% BSA in PBS during 1 hr at RT, followed by a blocking step of 30 min using 3% BSA in PBS. The cells were starved overnight in 9% serum replacement medium (SRM) composed of 46.5% AIM-V (Life Technologies, Madrid, Spain), 5% RPMI (Life Technologies) and 1% NEAA (Non-Essential Amino Acid Solution, Sigma-Aldrich) supplemented with 1% FN-depleted calf serum. $5 \times 10^4$ cells were plated, allowed to adhere for the indicated times and medium was removed and wells washed three times with PBS. The cells were stained with crystal violet (20% Methanol, 0.1% Crystal Violet) overnight at 4°C, washed, 0.1% triton X-100 was added and incubated for 2 hr at RT. Absorbance was measured at 595 nm.

## Spinning disk assay

The spinning disk assay was done as previously described (*Boettiger, 2007*) on POMA-treated glass coverslips of 25 mm diameter, coated with a solution of 10 µg/ml of purified pFN during 1 hr at RT and afterwards blocked 1 hr with 1% BSA in PBS. The *Fn1*-KO or HT1080 cells were starved overnight in 9% SRM supplemented with 1% FN-depleted serum. $7 \times 10^5$ cells were seeded, allowed to adhere for 1 hr and spun for 5 min at 6000 rpm in Dulbecco´s PBS supplemented with 80 mM $CaCl_2$ and 80 mM $MgCl_2$. After spinning the cells were fixed with 4% PFA and nuclei stained with DAPI. The nuclei were counted with a Zeiss Axiovert (objective 10x) controlled by Metamorph software, which allows taking images at determined positions. Data were analyzed as described (*Boettiger, 2007*). We calculated for each condition the τ50, which is the mean force for cell detachment.

## pFN-integrin crosslinking assay

Cells were seeded onto pFN-coated glass coverslips and spun and non-spun cells were incubated with 1 mM 3,3'-dithiobis (sulfosuccinimidyl propionate) (DTSSP; Thermo Scientific, Madrid, Spain) during 15 min at 4°C. Quenching was carried out with 50 mM Tris, pH 7.4 for 15 min at RT and cells were extracted with 20 mM Tris, pH 7.4, 0.1% SDS and proteinase inhibitors (Inhibitors cocktail, Roche , Barcelona, Spain). Cell lysates were collected and coverslips were thoroughly washed with 20 mM Tris, pH 8.5 followed by incubation with 20 mM Tris, pH 8.5, 0.1% SDS and 25 mM DTT for 1 hr at 37°C to break the crosslinks. The whole crosslinked fractions and the cell lysates were separated by SDS-PAGE and transferred to a nitrocellulose membrane. Western-blots were analyzed with ImageJ and the levels of crosslinked integrins were calculated as the relation between the crosslinked and the total integrin fractions (cell lysates + crosslinked fraction).

## Single-cell force spectroscopy (SCFS)

For cell attachment, cantilevers were plasma cleaned (PDC-32G, Harrick Plasma, Ithaca, NY, USA) and then incubated overnight at 4°C in PBS containing ConA (2 mg/ml, Sigma-Aldrich) (*Friedrichs et al., 2013*). For substrate coatings, 200 µm thick four-segmented polydimethylsilane (PDMS) mask fused to the surface of glass bottom Petri dishes (WPI, Sarasota, FL, USA) was used (*Yu et al., 2015*). Each of the four PDMS framed glass surfaces were incubated overnight at 4°C either with the FNIII7-10$^{wt}$ or FNIII7-10$^{syn}$ fragments or full-length FN (50 µg/ml) in PBS. For SCFS, we mounted an AFM (Nanowizard II, JPK Instruments, Berlin, Germany) on an inverted fluorescence microscope (*Puech et al., 2006*) (Observer Z1/A1, Zeiss, Germany). The temperature was kept at 37°C throughout the experiment by a Petri dish heater (JPK Instruments,Berlin, Germany). 200 µm long tip-less V-shaped silicon nitride cantilevers having nominal spring constants of 0.06 N/m (NP-0, Bruker, USA) were used. Each cantilever was calibrated prior the measurement by determining its sensitivity and spring constant using the thermal noise analysis of the AFM (*Hutter and Bechhoefer, 1993*). To adhere a single fibroblast to the AFM cantilever, overnight serum-starved fibroblasts with confluency up to ≈80% were washed with PBS, trypsin-detached for up to 2 min, suspended in SCFS media (DMEM supplemented with 20 mM HEPES) containing 1% (v/v) FCS, pelleted and resuspended in serum free SCFS media. Fibroblasts were allowed to recover for at least 30 min from trypsin treatment. Adhesion of a single fibroblast to the free cantilever end was achieved by pipetting the fibroblast suspension onto the functionalized Petri dishes. The functionalized cantilever was lowered onto a fibroblast with a speed of 1 µm/s until a force of 1 nN was recorded. After ≈5 s contact, the cantilever was retracted with 1 µm/s for 10 µm and the cantilever bound fibroblast was incubated for 7–10 min to assure firm binding to the cantilever. Using differential interference contract (DIC) microscopy, the morphological state of the fibroblast was monitored. For single molecule sensitivity, the fibroblast bound to the cantilever was lowered onto the coated substrate with a speed of 1 µm/s until a contact force of 200 pN was recorded for ≈50 ms contact time. Subsequently, the cantilever was retracted at 1 µm/s and for ≥13 µm until the fibroblast and substrate were fully separated. After the experimental cycle, the fibroblast was allowed to recover for 0.5 s. For each measurement, the area of the substrate was varied. Force-distance curves were analyzed to determine binding probability using JPK software. Mann-Whitney tests were applied to determine significant differences between the binding probability of fibroblast lines at different conditions. Tests were done using Prism (GraphPad, La Jolla, USA).

## pFAK analysis

Cells were plated on pFN-coated glass coverslips and spun in the spinning disk device, then lysed in RIPA buffer (50 mM Tris, pH 7.4; 1% NP-40; 0.5% Na-Deoxycolate; 0.1% SDS; 2 mM EDTA) supplemented with proteinase inhibitors (Complete Proteinase Inhibitor Cocktail tablet, Roche), phosphatase inhibitors (Protease Inhibitors Cocktail 2 Aqueous Solution and Cocktail 3, Sigma-Aldrich), 1 mM $Na_3VO_4$ and 5 mM NaF for 10 min on ice, and sonicated for 1 min. The protein concentrations were quantified using the Pierce BCA Protein Assay Kit (Thermo Scientific) assay and 30–50 μg of protein were separated by SDS-PAGE gel, transferred to nitrocellulose membranes and hybridized with specific antibodies. Western-blots were analyzed with ImageJ and the levels of phospho-Tyr$_{397}$-FAK or phospho-Tyr$_{861}$FAK were referred to the total FAK content.

## FN matrix assembly assay

*Fn1*-KO fibroblasts were starved overnight in 9% SRM supplemented with 1% FN-depleted serum, trypsinized and transferred into 8-well Lab-Tek chambers (Thermo Scientific) coated for 1 hr with a solution of 20 μg/ml of Laminin (Roche) at RT. After 3 hr, the 9% SRM was supplemented with 1% mouse plasma and cells were incubated for 24, 48, 72 and 96 hr, fixed with 4% PFA and prepared for immunofluorescence staining.

## Cell spreading assay

Glass coverslips (18 × 18 mm) were POMA treated, coated with pFN and then incubated with $2 \times 10^4$ *Fn1*-KO or pKO-$\beta$1, pKO-$\alpha$v and pKO-$\alpha$v/$\beta$1 cells starved overnight in 9% SRM supplemented with 1% FN-depleted serum. After 15, 30, 60 and 120 min of adhesion, cells were fixed with 2% PFA and immunostained. Focal adhesions were quantified with imageJ.

## Integrin expression analysis by FACS

Flow cytometry to analyse integrin levels on the *Fn1*-KO fibroblasts was carried out as previously described (*Theodosiou et al., 2016*).

## Live imaging of cell spreading

$10^4$ cells (*Fn1*-KO, pKO-$\beta$1, pKO-$\alpha$v or pKO-$\alpha$v/$\beta$1) were starved overnight, cultured on μ-Slide eight well chambers (Ibidi, Martinsried, Munich) coated with 10 μg/ml of pFN during 1 hr and imaged with frame rates of 90 s in a Zeiss Axiovert microscope using the VisiView (Visitron Systems, Puchheim, Germany) software.

## Histological analysis

Adult mice were perfused with 4% parafolmaldehyde (PFA) in PBS or tissue pieces and embryos were fixed overnight with 4% PFA at 4°C. Fixed tissues were dehydrated in graded alcohol series, embedded in paraffin (Paraplast X-tra, Sigma-Aldrich), sectioned into 8 μm thick sections and stained with Haematoxylin-Eosin (H and E) using standard protocols. For immunostainings, sections were hydrated with inverse graded alcohol series, unmasked by heating in 10 mM citrate buffer (pH 6) for 10 min, blocked for 1 hr with 3% BSA at RT and incubated overnight with the primary antibody, washed, incubated with secondary antibodies for 1 hr at room temperature, washed, DAPI stained and mounted on glass slides with elvanol.

## Embryo and skin whole mount immunostaining

Embryos were isolated from pregnant mothers at the stages of E11.5, E15.5 and E16.5 and fixed overnight at 4°C with DENT´s fixative consisting of 80% Methanol, 20% DMSO. The skin was dissected after fixation from the E15.5 and E16.5 embryos, washed 3 times with 100% methanol (5 min) and kept at −20°C in 100% methanol. For staining, fixed pieces of skin or whole E11.5 embryos were hydrated in decreasing (75, 50 and 25%) methanol series, diluted in PBS supplemented with 0.1% Tween20 (PBST) and blocked for 2 hr at RT with 3% BSA in PBST. Incubations with primary and secondary antibodies were done overnight at 4°C with gentle rocking and after washing with PBST, tissues were mounted with elvanol.

## Platelet isolation and quantification

Blood from *Fn1^syn/syn* or *Fn1^+/+* mice was collected in heparinized Microvette CB 300 LH tubes (Sarstedt) and platelets were counted using a ProCyte Hematology Analyzer (IDEXX Laboratories, Ludwigsburg, Germany). To isolate platelets, heparinized blood from $\beta3^{+/+}$ or $\beta3^{-/-}$ mice was centrifuged at 70xg for 10 min at RT, the platelet enriched upper phase was then centrifuged at 800xg for 10 min and the platelet pellet was finally washed twice with Tyrodes buffer pH 6.5 (134 mM NaCl, 2.9 mM KCl, 12 mM NaHCO$_3$, 10 mM *N*-2-hydroxyethylpiperazine-*N*-2-ethanesulfonic acid, 5 mM glucose, 0.35% bovine serum albumin (BSA)). Washed platelets were resuspended in Tyrodes buffer pH 7.4 and counted using a ProCyte Hematology Analyzer (IDEXX Laboratories). For experiments, platelet numbers were adjusted to equivalent concentrations with Tyrodes buffer pH 7.4 complemented with 1 mM CaCl$_2$, 1 mM MgCl$_2$.

## Platelet aggregation in vitro assays

Platelet aggregation was measured with $1 \times 10^8$ washed platelets stimulated with 0.5 U/ml thrombin (Sigma-Aldrich) or 5 µg/ml fibrillar type I collagen (Nycomed, Munich, Germany) in the presence of 10 µg/ml pFN isolated either from *Fn1^+/+* or *Fn1^syn/syn* mice. For platelet aggregation with 20 µM ADP, platelet rich plasma (PRP) was isolated. The mouse blood was collected with citrate buffer (1:9, buffer:blood), centrifuged at 110xg and the supernatant (PRP) was collected. A volume of 225 µl of PRP containing $6.75 \times 10^7$ platelets was used for each experiment adding 20 µM ADP. Light transmission was recorded with a ChronoLog aggregometer over 15 min as arbitrary units with the transmission through buffer defined as 100% transmission.

## FN and fibrinogen quantification in isolated platelets and blood plasma

Platelets were isolated from *Fn1^+/+* and *Fn1^syn/syn* heparinized blood as described above. About $5 \times 10^6$ platelets were lysed with 0,1% Triton in TBS with proteinase inhibitors (Complete Proteinase Inhibitor Cocktail tablet, Roche) during 10 min on ice. After centrifugation at 13,000 rpm, the supernatant was run in an 8% SDS-PAGE under reducing conditions, transferred to nitrocellulose membranes and incubated with anti-FN antibodies. To quantify the plasma content of FN and fibrinogen, 2 µl of plasma were loaded onto the 8% SDS-PAGE. As a reference, we used pure human pFN (Millipore) and human fibrinogen (Sigma-Aldrich). Western-blots were analyzed with ImageJ. To know the FN levels in platelets derived from the different mouse strains the FN levels were related to their vinculin contents.

## Platelet spreading assay

To study platelet spreading, glass bottom dishes were coated with 10 µg/ml of pFN^wt, pFN^syn, fibrinogen (Sigma-Aldrich) or soluble collagen type I (PureCol, Advanced Biomatrix, San Diego, CA, USA) at RT for 1 hr and blocked with 1% BSA in PBS. Washed platelets ($0.5–1 \times 10^6$) were added to the dishes in a final volume of 1 ml and activated with 0.01% thrombin (Sigma-Aldrich). Images were taken after 15, 30 and 60 min under a differential interference contrast microscopy (Zeiss Axiovert 200M microscope with a Plan-NEOFLUAR,×100, 1.45 oil objective; Zeiss, Jena, Germany) using the Metamorph software (Molecular Devices, Sunnyvale, CA, USA). The platelet spreading area was analysed using the ImageJ software.

## Platelets adhesion assay under flow

Flow chamber experiments were carried out as described previously (*Schulz et al., 2009*) using the air-driven continuous flow pump system from Ibidi. Briefly, platelets were isolated, fluorescently labelled with 5 µM carboxyfluorescein succinimidyl ester (CFSE; Invitrogen) in Tyrodes buffer pH 6.5 for 15 min and then washed. To achieve near-physiological conditions during perfusion of the pFN-coated flow chamber slides, 2 ml of washed platelets with a platelet count of $1 \times 10^7$ were combined with 1 ml of human erythrocytes isolated from the blood of a healthy volunteer.

For each experiment, 4 channels of a flow chamber slide (µ-Slide VI 0.1 ibiTreat, Ibidi) were coated with 10 µg/ml fibrillar collagen, fibrinogen, pFN^wt or pFN^syn over night at 4°C and blocked with 1% BSA the following day. The coated channels of one µ-slide were connected in series with connector tubings for simultaneous perfusion. The platelet suspension was filled in one reservoir of a Perfusion Set Black (Ibidi) and the pump was started with unidirectional flow at the highest possible

pressure 100 mbar) until all channels were filled with the blood-like fluid. Then, the experiment was started by adjusting the shear rate to approximately 1000/s. The channels were perfused for 10 min and subsequently washed by perfusing Tyrodes buffer for another 10 min. Platelets were imaged after performing the perfusion with a Zeiss Apotome microscope and platelet surface coverage was analysed using ImageJ.

## Microvascular thrombus formation

The surgical preparation of the mouse cremaster muscle was performed as described (*Baez, 1973*). Mice were anesthetized using a mixture of 100 mg/kg ketamine and 10 mg/kg xylazin. The left femoral artery was cannulated in a retrograde manner to administer FITC-labeled dextran (MW 150 kDa; Sigma Aldrich). The right cremaster muscle was exposed through a ventral incision of the scrotum. The muscle was opened ventrally in a relatively avascular zone and spread over the pedestal of a custom-made microscopy stage. Epididymis and testicle were detached from the cremaster muscle and placed into the abdominal cavity. Throughout the surgical procedure and in vivo microscopy, the muscle was superfused with warm saline solution. At the end of each experiment, blood samples were collected by cardiac puncture to determine systemic cell counts using a hematology analysis system (ProCyte DX, IDEXX Laboratories ).

Microvascular thrombus formation was induced by phototoxic injury as described (*Rumbaut et al., 2005*) with slight modifications. Briefly, after surgical preparation of the cremaster muscle, 4 ml/kg body weight of a 2.5% solution of FITC-dextran was infused intraarterially and the exposed the vessel segment under investigation was continuously epi-illuminated with a wavelength of 488 nm (Polychrome II, TILL Photonics, Gräfelfing, Germany). An Olympus water immersion lens (60 × /NA 0.9) in an upright microscope (BX50; Olympus Microscopy, Hamburg, Germany) was used to focus the light onto the cremaster muscle and to visualize the microvascular thrombus formation in real-time. Thrombus formation was induced in one arteriole (25–35 µm) per experiment by analyzing the time until the first platelet adhesion to the vessel wall (defined as the onset of thrombus formation) occurred and the time until blood flow ceased (defined as the complete occlusion of the vessel).

## Tail bleeding assay

The tail bleeding assay was performed in anesthetized mice directly after the analysis of microvascular thrombus formation. For this purpose, the distal 2 mm segment of the tail was removed with a scalpel. Bleeding was monitored by absorbing the bead of blood with a filter paper in 30 s intervals without touching the wound. Tail bleeding time was defined as the time until hemorrhage ceased.

## Acknowledgements

We acknowledge support from the Spanish Ministry for Economy and Competitiveness (MINECO) and Fondo Europeo de Desarrollo Regional (FEDER): Mat2012-38359 (MINECO) and Mat2015-69315 (MINECO/FEDER) as well as from the Sonderforschungsbereich 914 (SFB 914; project B3) granted by the Deutsche Forschungsgemeinschaft (DFG). MB-J was supported by a contract from the Conselleria Valenciana d'Educació i Ciència. We thank Kairbaan Hodivala-Dilke for providing the *Itgb3-/-* mouse strain and Herbert Schiller for cell lines.

## Additional information

### Funding

| Funder | Grant reference number | Author |
| --- | --- | --- |
| Ministerio de Economía y Competitividad | National grant | Maria Benito-Jardón Irene Gimeno-LLuch Mercedes Costell |
| Conselleria Valenciana d'Educació i Ciència | Graduate student fellowship | Maria Benito-Jardón |

The funders had no role in study design, data collection and interpretation, or the decision to submit the work for publication.

## Author contributions

MB-J, Formal analysis, Investigation, Writing—review and editing; SK, MB, Investigation, Writing—review and editing; IG-L, Resources, Investigation, Writing—review and editing; TP, Conceptualization, Formal analysis, Writing—review and editing; DJM, Conceptualization, Supervision, Investigation, Writing—original draft; GZ, Investigation, Writing—original draft; CAR, Conceptualization, Supervision, Writing—original draft; MC, Conceptualization, Resources, Supervision, Funding acquisition, Validation, Investigation, Visualization, Methodology, Writing—original draft, Project administration, Writing—review and editing

## Author ORCIDs

Maria Benito-Jardón, http://orcid.org/0000-0001-9562-5430
Mercedes Costell, http://orcid.org/0000-0001-6146-996X

## Ethics

Animal experimentation: Mice were housed in special pathogen free animal facilities. All mouse work was performed in accordance with the Government of the Valencian Community (Spain) guidelines (permission reference A1327395471346). Mice containing the integrin β3 deletion were bred under the permission reference 55.2-1-54-2532-96-2015 (Government of Upper Bavaria). The tail-bleeding and cremaster muscle venules injury assays performed under the permission reference 55.2-1-54-2532-115-12 (Government of Upper Bavaria).

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
