## [Decision Letter]

Thank you for submitting your article "The fibronectin synergy site re-enforces cell adhesion and mediates a crosstalk between integrin classes" for consideration by *eLife*. Your article has been favorably evaluated by Fiona Watt as the Senior Editor and three reviewers, including Ambra Pozzi (Reviewer #2), Heyu Ni (Reviewer #3), and a member of our Board of Reviewing Editors.

The reviewers have discussed the reviews with one another and the Reviewing Editor has drafted this decision to help you prepare a revised submission.

Summary:

You and your coworkers describe the functional properties of the FN synergy site in mice and cells. You report (1) that the absence of the synergy site produces normal offspring that have only a mild bleeding tendency upon wounding; (2) that intercrosses with *Itgb3*-null mice lead to lethal bleeding due to dysfunctional platelets and mural cells; (3) that plasma FN isolated from mutant mice supports adhesion strengthening rather than an increased on-rate/affinity of α5/β1 for FN. The reviewers felt that interesting and the data are of high quality. However, they raised questions that should be addressed prior to publication.

Essential revisions:

1) The paper would benefit if the authors discuss and compare defects of FN-synergy site-deficient mice and mice carrying an RGD>RGE substitution. This is lacking in the manuscript. The latter mice lack α5/β1 binding and develop defects in paraxial mesoderm and lack tail-bud-derived somites leading to a shortened posterior trunk indicating that the FN-synergy site dependent adhesion re-enforcement is not required for the reported compensation of α5 and αv integrins during (paraxial) mesoderm formation and posterior somitogenesis.

2) The authors conducted all in vitro experiments with purified FN from wild type and FN-synergy site-deficient mice, except the AFM studies. It would be interesting to also test α5 on rate/activation with full length FN molecules.

3) In Figure 1, the tail bleeding time in *FN^syn/syn^*mice was observed to be significantly prolonged as compared to WT mice. However, while FN synergy site may be important for hemostasis, it was reported that tail bleeding time (a less than perfect parameter for hemostasis itself) in plasma FN conditional knockout mice was not prolonged (Sakai T et al., Nat Med, 2001). It seems that prolonged bleeding time is found in plasma *FN^-/-^* mice only with combined deficiency of fibrinogen or following anti-coagulant treatment (Wang Y et al., J Clin Invest, 2014). Are there additional defects in subendothelial matrix organization in *FN^syn/syn^*mice? Further discussions or experiments are required to explain this bleeding time prolongation in *FN^syn/syn^*mice.

4) There is no information regarding the platelet aggregation in platelet-rich plasma induced by ADP, which is the most common platelet aggregation assay and important information for hemostasis. The concentration of plasma FN in *FN^syn/syn^*mice is also unclear. These experiments are very easy to do and the information is important for readers to better understand the bleeding disorder of *FN^syn/syn^*mice.

The authors should be aware that plasma FN is an inhibitory factor for platelet aggregation no matter if it is in plasma or released from platelets that internalized by αIIbβ3 from plasma (Reheman A et al., Blood. 2009). However, plasma FN can enhance platelet aggregation when it is associated with fibrin. The authors should read these earlier publications and discuss whether the 70% FN reduction in *FN^syn/syn^*platelets is the explanation of the trends of enhancement of platelet aggregation in Figure 1—figure supplement 1).

5) Figure 3 showed a significant decrease of the adhesion strength, indicated by inflection point (τ50) on FN^syn^ coated surface. However, this result is inconsistent with the curves shown in Figure 3—figure supplement 1, where the τ50 is clearly higher on FN^syn^ coated surface than on WT FN coated surface.

6) Introduction (first paragraph): It has been now well known that αIIbβ3 is not "exclusively expressed on platelets". It is not only expressed on platelet precursor megakaryocytes, but also mast cells and hematopoietic stem cells. This sentence should be corrected/revised.

[Editors' note: further revisions were requested prior to acceptance, as described below.]

Thank you for resubmitting your work entitled "The fibronectin synergy site re-enforces cell adhesion and mediates a crosstalk between integrin classes" for further consideration at *eLife*. Your revised article has been favorably evaluated by Fiona Watt as the Senior Editor and three reviewers, one of whom is a member of our Board of Reviewing Editors.

The manuscript has been improved but there are some remaining issues (regarding editing cartoons and text) that need to be addressed before acceptance, as outlined below:

1) Figure 6: the authors indicate that the 'intensity' of the color in their cartoon in Figure 6 indicates whether integrins are activated or not. However, in the panel to the right of the cartoon they provide color intensities that are not depicted in the cartoon on the left. Furthermore, the authors need to clearly indicate that darker colors indicate more active integrins than light colors. Finally, the difference in color and/or shape (if any) between integrins activated with or without synergy engagement is not clear.

2) Figure 8: although the figure is very nice and informative, it would increase clarity if the authors change the color of integrin α2β1 (green), which is quite difficult to see on the blue and green background of the platelets. Furthermore, fibrinogen and importantly fibrin should be easily visible in the platelet aggregates, to avoid giving the readers a wrong impression that FN alone is sufficient to mediate platelet aggregation. Fibrin is found to be required for FN to support platelet aggregation (Reheman A, Blood, 2009; Wang Y, JCI, 2014). Another important fact is that FN and fibrin fibrils are more abundant close to the vessel wall.

3) In platelet aggregation tracings in Figure 1—figure supplement 2, the values obtained from *Fn1^syn/syn^* were "shifted by +5% to avoid overlapping". This is not standard presentation and may mislead the readers. We recommend the authors to allow overlapping of aggregation tracings in the figure, particularly when they used different colors to show their aggregation curves.

4) Please correct the following typos: Results: "comparable normal" should be changed to "comparable to Fn1^+/+^ mice and normal"; Subsection “Normal development and prolonged trauma-induced bleeding in *Fn1^syn/syn^* mice” "also" should not be at the beginning of the sentence but rather after "were"; Subsection “The FN synergy site is dispensable for the on-rate of FN binding to α5β1 integrins”, last paragraph: "site" is missing after "synergy"; Discussion, fourth paragraph: it should be acknowledged that platelet adhesion to Fn is mainly mediated by αIIbβ3 integrin under normal conditions. The authors may consider whether they modify this sentence to: It is well known that platelet adhesion to pFN, mediated by αIIbβ3 with contributions from α5β1 and other integrins, plays a critical role for hemostasis.

---

## [Author Response]

*Essential revisions:*

*1) The paper would benefit if the authors discuss and compare defects of FN-synergy site-deficient mice and mice carrying an RGD>RGE substitution. This is lacking in the manuscript. The latter mice lack α5/β1 binding and develop defects in paraxial mesoderm and lack tail-bud-derived somites leading to a shortened posterior trunk indicating that the FN-synergy site dependent adhesion re-enforcement is not required for the reported compensation of α5 and αv integrins during (paraxial) mesoderm formation and posterior somitogenesis.*

This is a very important point that is still not fully resolved. The α5/αv integrin double knockout mice die at around E8.5 with severe mesoderm defects and absent somites. The FN^RGE^ mice die at E9.5-10 with the same phenotype as α5 integrin-null mice and indicating that the RGD>RGE mutation impairs α5 binding to FN and that the synergy site does not suffice α5-mediated adhesion. We added this point to the discussion.

The second important point is whether the synergy site compensates the lack of αv- mediated adhesion to FN. The αv-null mice have no mesoderm defects. In our in vitro assays, the FN synergy site can compensate the absence of αvβ3 integrins. Unfortunately, we do not know yet whether this is also the case in vivo. We are currently crossing αv-null mice with FN^syn^ mice to answer this question. However these experiments will take at least one more year.

*2) The authors conducted all* in vitro *experiments with purified FN from wild type and FN-synergy site-deficient mice, except the AFM studies. It would be interesting to also test α5 on rate/activation with full length FN molecules.*

We did the experiments and found no significant differences between FN^wt^ and FN^syn.^These findingsindicate that the binding on-rate seems to be independent of the synergy site. We show this result in Figure 4. Interestingly the pKO cells that do not express integrins showed higher probability of association to the full length FN than to the FNIII7-10 fragment, pointing to non-integrin specific binding events to full length FN. Such binding events could for example be mediated by syndecans.

*3) In Figure 1, the tail bleeding time in FN^syn/syn^mice was observed to be significantly prolonged as compared to WT mice. However, while FN synergy site may be important for hemostasis, it was reported that tail bleeding time (a less than perfect parameter for hemostasis itself) in plasma FN conditional knockout mice was not prolonged (Sakai T et al., Nat Med, 2001). It seems that prolonged bleeding time is found in plasma FN^-/-^ mice only with combined deficiency of fibrinogen or following anti-coagulant treatment (Wang Y et al., J Clin Invest, 2014). Are there additional defects in subendothelial matrix organization in FN^syn/syn^mice? Further discussions or experiments are required to explain this bleeding time prolongation in FN^syn/syn^mice.*

We compared the subendothelial distributions of FN, laminin and collagen IV in dermal vasculature of control and *FN^syn/syn^*mice and found no differences. The results are shown in Figure 1—figure supplement 1. We also examined blood vessel ruptures by immunostaining erythrocytes. These results are shown in Figure 1—figure supplement 1.

It should be noted that protein mutations and absence of proteins can have different outcomes. The absence of pFN can trigger compensations that are not induced when pFN^syn^ is expressed. It could well be that the absence of pFN is compensated by the increased use of fibrinogen or other hemostatic proteins and that this not occur in *FN^syn/syn^*mice. We mention such a possibility in the revised Discussion.

*4) There is no information regarding the platelet aggregation in platelet-rich plasma induced by ADP, which is the most common platelet aggregation assay and important information for hemostasis. The concentration of plasma FN in FN^syn/syn^mice is also unclear. These experiments are very easy to do and the information is important for readers to better understand the bleeding disorder of FN^syn/syn^mice.*

*The authors should be aware that plasma FN is an inhibitory factor for platelet aggregation no matter if it is in plasma or released from platelets that internalized by αIIbβ3 from plasma (Reheman A et al., Blood. 2009). However, plasma FN can enhance platelet aggregation when it is associated with fibrin. The authors should read these earlier publications and discuss whether the 70% FN reduction in FN^syn/syn^platelets is the explanation of the trends of enhancement of platelet aggregation in Figure 1—figure supplement 1).*

We have done ADP-induced platelet aggregation assays in presence of plasma and observed no significant differences between wild-type and *FN^syn/syn^*mice plasma (shown in Figure 1—figure supplement 2). Furthermore, we quantified the concentration of plasma FN and fibrinogen in wild-type and *FN^syn/syn^*mice and measured 318.7 and 316.1 μg/ml of plasma FN and 2.10 and 2.08 mg/ml of fibrinogen respectively. The findings are displayed in the Results and in Figure 1—figure supplement 2.

The platelet aggregations showed in Figure 1—figure supplement 1 were not significant. The figures appear enhanced in the FNsyn condition because to avoid overlapping of the graphs, 5 units were added to every resulting value in the aggregations with FN^syn^. Now we explain this in the legend of the figure (currently Figure 1—figure supplement 2).

We repeated the aggregation assays (the three conditions: with collagen, with thrombin and with ADP) and just do not find significant differences between the wild-type and mutant genotypes. The decreased content of pFN in the platelet α-granules from FN^syn^ mice indicates that αIIbβ3 integrin adhesion to pFN^syn^ is not strong enough to allow an efficient uptake of pFN but its reduction is possibly masked by the high amount of FN in plasma.

*5) Figure 3 showed a significant decrease of the adhesion strength, indicated by inflection point (τ50) on FN^syn^ coated surface. However, this result is inconsistent with the curves shown in Figure 3—figure supplement 1, where the τ50 is clearly higher on FN^syn^ coated surface than on WT FN coated surface.*

We made a mistake in the legend of the Figure 3—figure supplement 1. We apologize and corrected the mistake.

*6) Introduction (first paragraph): It has been now well known that αIIbβ3 is not "exclusively expressed on platelets". It is not only expressed on platelet precursor megakaryocytes, but also mast cells and hematopoietic stem cells. This sentence should be corrected/revised.*

We thank to the referees for this note. We corrected the statement.

[Editors' note: further revisions were requested prior to acceptance, as described below.]

*The manuscript has been improved but there are some remaining issues (regarding editing cartoons and text) that need to be addressed before acceptance, as outlined below:*

*1) Figure 6: the authors indicate that the 'intensity' of the color in their cartoon in Figure 6 indicates whether integrins are activated or not. However, in the panel to the right of the cartoon they provide color intensities that are not depicted in the cartoon on the left. Furthermore, the authors need to clearly indicate that darker colors indicate more active integrins than light colors. Finally, the difference in color and/or shape (if any) between integrins activated with or without synergy engagement is not clear.*

*2) Figure 8: although the figure is very nice and informative, it would increase clarity if the authors change the color of integrin α2β1 (green), which is quite difficult to see on the blue and green background of the platelets. Furthermore, fibrinogen and importantly fibrin should be easily visible in the platelet aggregates, to avoid giving the readers a wrong impression that FN alone is sufficient to mediate platelet aggregation. Fibrin is found to be required for FN to support platelet aggregation (Reheman A, Blood, 2009; Wang Y, JCI, 2014). Another important fact is that FN and fibrin fibrils are more abundant close to the vessel wall.*

*3) In platelet aggregation tracings in Figure 1—figure supplement 2, the values obtained from Fn1^syn/syn^ were "shifted by +5% to avoid overlapping". This is not standard presentation and may mislead the readers. We recommend the authors to allow overlapping of aggregation tracings in the figure, particularly when they used different colors to show their aggregation curves.*

4) Please correct the following typos: Results: "comparable normal" should be changed to "comparable to Fn1^+/+^ mice and normal"; Subsection “Normal development and prolonged trauma-induced bleeding in Fn1^syn/syn^ mice” "also" should not be at the beginning of the sentence but rather after "were"; Subsection “The FN synergy site is dispensable for the on-rate of FN binding to α5β1 integrins”, last paragraph: "site" is missing after "synergy"; Discussion, fourth paragraph: it should be acknowledged that platelet adhesion to Fn is mainly mediated by αIIbβ3 integrin under normal conditions. The authors may consider whether they modify this sentence to: It is well known that platelet adhesion to pFN, mediated by αIIbβ3 with contributions from α5β1 and other integrins, plays a critical role for hemostasis.

We agree that the reviewers’ suggestions will improve the clarity of our cartoons. We made all the requested changes in the figures and corrected the English in the sentences listed by the reviewers. Finally, we adapted the legends of those figures that were corrected according to the reviewers’ suggestions and rephrased the last sentence in the Discussion.